# SHLD2/FAM35A co-operates with REV7 to coordinate DNA double-strand break repair pathway choice

Steven Findlay[1,2,†], John Heath[1,2,†], Vincent M Luo[1,3], Abba Malina[1,4] (ID), Théo Morin[5], Yan Coulombe[6,7], Billel Djerir[5], Zhigang Li[1], Arash Samiei[1,2], Estelle Simo-Cheyou[1,4], Martin Karam[2], Halil Bagci[8,9], Dolev Rahat[10], Damien Grapton[1], Elise G Lavoie[1], Christian Dove[1,2], Husam Khaled[1,2], Hellen Kuasne[11], Koren K Mann[1,2,4], Kathleen Oros Klein[1], Celia M Greenwood[1,12], Yuval Tabach[10], Morag Park[11], Jean-Francois Côté[8,9,13,14] (ID), Jean-Yves Masson[6,7], Alexandre Maréchal[5] & Alexandre Orthwein[1,2,3,4,*] (ID)

## Abstract

DNA double-strand breaks (DSBs) can be repaired by two major pathways: non-homologous end-joining (NHEJ) and homologous recombination (HR). DNA repair pathway choice is governed by the opposing activities of 53BP1, in complex with its effectors RIF1 and REV7, and BRCA1. However, it remains unknown how the 53BP1/RIF1/REV7 complex stimulates NHEJ and restricts HR to the S/G2 phases of the cell cycle. Using a mass spectrometry (MS)-based approach, we identify 11 high-confidence REV7 interactors and elucidate the role of SHLD2 (previously annotated as FAM35A and RINN2) as an effector of REV7 in the NHEJ pathway. FAM35A depletion impairs NHEJ-mediated DNA repair and compromises antibody diversification by class switch recombination (CSR) in B cells. FAM35A accumulates at DSBs in a 53BP1-, RIF1-, and REV7-dependent manner and antagonizes HR by limiting DNA end resection. In fact, FAM35A is part of a larger complex composed of REV7 and SHLD1 (previously annotated as C20orf196 and RINN3), which promotes NHEJ and limits HR. Together, these results establish SHLD2 as a novel effector of REV7 in controlling the decision-making process during DSB repair.

**Keywords** DNA double-strand break; DNA repair pathway choice; non-homologous end-joining; REV7

**Subject Categories** DNA Replication, Repair & Recombination
The EMBO Journal (2018) 37: e100158

## Introduction

Due to their highly recombinogenic and pro-apoptotic potential, DNA double-strand breaks (DSBs) are one of the most cytotoxic DNA lesions. Their inaccurate resolution can result in point mutations, small deletions/insertions, chromosomal rearrangements, or loss of gross genetic information that drive genomic instability, carcinogenesis, and cell death (reviewed in Tubbs & Nussenzweig, 2017). To avoid these deleterious outcomes, cells have deployed a complex network of proteins to signal and repair DSBs. One critical step during the DSB response consists in the choice between two mutually exclusive DNA repair pathways: non-homologous end-joining (NHEJ) and homologous recombination (HR; reviewed in Ceccaldi *et al*, 2016). This decision process, named DNA repair pathway choice, integrates several elements including the cell cycle status, the complexity of the DNA end, and the epigenetic context. Importantly, DNA repair pathway choice is under the control of two

1 Lady Davis Institute for Medical Research, Segal Cancer Centre, Jewish General Hospital, Montreal, QC, Canada
2 Division of Experimental Medicine, McGill University, Montreal, QC, Canada
3 Department of Microbiology and Immunology, McGill University, Montreal, QC, Canada
4 Gerald Bronfman Department of Oncology, McGill University, Montreal, QC, Canada
5 Department of Biology, Université de Sherbrooke, Sherbrooke, QC, Canada
6 Genome Stability Laboratory, CHU de Québec Research Center, Quebec City, QC, Canada
7 Department of Molecular Biology, Medical Biochemistry and Pathology, Laval University Cancer Research Center, Quebec City, QC, Canada
8 Institut de Recherches Cliniques de Montréal (IRCM), Montreal, QC, Canada
9 Department of Anatomy and Cell Biology, McGill University, Montreal, QC, Canada
10 Department of Developmental Biology and Cancer Research, The Institute for Medical Research Israel-Canada, The Hebrew University of Jerusalem, Jerusalem, Israel
11 Rosalind and Morris Goodman Cancer Research Centre, McGill University, Montreal, QC, Canada
12 Department of Epidemiology, Biostatistics and Occupational Health, MGill University, Montreal, QC, Canada
13 Département de Biochimie et Médecine Moléculaire, Université de Montréal, Montreal, QC, Canada
14 Département de Médecine (Programmes de Biologie Moléculaire), Université de Montréal, Montreal, QC, Canada
*Corresponding author. Tel: +1 514 340 8222; E-mail: alexandre.orthwein@mcgill.ca
†These authors contributed equally to this work

antagonizing factors, 53BP1 and BRCA1 (reviewed in Hustedt & Durocher, 2016).

Non-homologous end-joining is predominantly involved in the repair of DSBs during the G1 phase of the cell cycle. It is characterized by a limited processing of the DNA ends catalyzed by the nuclease Artemis and their subsequent ligation by DNA ligase IV (reviewed in Betermier *et al*, 2014). Importantly, NHEJ is promoted by the recruitment of 53BP1 at DSBs, along with its effectors RIF1, REV7, and PTIP (Chapman *et al*, 2012, 2013; Callen *et al*, 2013; Di Virgilio *et al*, 2013; Escribano-Diaz *et al*, 2013; Feng *et al*, 2013; Zimmermann *et al*, 2013; Boersma *et al*, 2015; Xu *et al*, 2015). These latter factors play a central role in several additional biological processes, including the establishment of a protective immunity during class switch recombination (CSR), a programmed DSB-dependent process that specifically occurs in B cells (Manis *et al*, 2004; Ward *et al*, 2004; Chapman *et al*, 2013; Di Virgilio *et al*, 2013; Escribano-Diaz *et al*, 2013; Boersma *et al*, 2015; Xu *et al*, 2015).

In S/G2 phases of the cell cycle (when sister chromatids are available as templates), HR is activated and can alternatively repair DSBs. One of the key features of HR is the formation of long stretches of single-stranded DNA (ssDNA), a process called DNA end resection (reviewed in Fradet-Turcotte *et al*, 2016). The resulting ssDNA stretches are rapidly coated by RPA, which is subsequently replaced by the recombinase RAD51 to form nucleofilaments that are a pre-requisite for the subsequent search of homology, strand invasion, and strand exchange before the resolution of the DSB by the HR machinery. Critically, BRCA1 promotes the initiation of DNA end resection and HR-mediated DSB repair by preventing the recruitment of 53BP1 and its downstream effectors to sites of DNA damage in S/G2 phases (Chapman *et al*, 2012, 2013; Escribano-Diaz *et al*, 2013; Feng *et al*, 2013), thereby antagonizing 53BP1 function in NHEJ.

While the opposing role of 53BP1 and BRCA1 in DNA repair pathway choice has been extensively scrutinized over the past years, it remains largely unclear how the 53BP1 downstream effectors, namely REV7, promote NHEJ and antagonize BRCA1-mediated HR in G1 phase of the cell cycle (Boersma *et al*, 2015; Xu *et al*, 2015). REV7 is an adaptor protein that has been described for its role in mitotic progression through the control of both the activity of the spindle assembly checkpoint (SAC) and the formation of a functional anaphase-promoting complex/cyclosome-Cdc20 (APC/C; Cahill *et al*, 1999; Listovsky & Sale, 2013; Bhat *et al*, 2015). In parallel, REV7 is a well-defined player in DNA translesion synthesis (TLS; reviewed in Waters *et al*, 2009) as well as DSB repair by HR as part of a complex composed of the deoxycytidyl (dCMP) transferase REV1 and the catalytic subunit of the DNA polymerase ζ, REV3L (Sharma *et al*, 2012). The recent discovery that REV7 participates in the NHEJ pathway in a TLS-independent manner raised fundamental questions about how this adaptor protein promotes DSB repair and controls DNA repair pathway choice.

In this present study, we sought to get insight into the decision-making process underpinning DNA repair pathway choice by deciphering the interactome of REV7. Using a mass spectrometry (MS)-based approach, we identified SHLD2 (previously known as FAM35A/RINN2) as an effector of REV7 in the NHEJ pathway. FAM35A accumulates at DSBs through its N-terminal domain in a 53BP1-, RIF1-, and REV7-dependent manner. Importantly, depletion of FAM35A impairs both NHEJ and CSR, while promoting DNA end resection and HR. In fact, FAM35A acts in concert with SHLD1 (previously known as C20orf196/RINN3) in promoting both NHEJ and CSR while antagonizing HR. Altogether, our results provide a better insight into the molecular events that control DNA repair pathway choice.

# Results

## Mapping of REV7 proximal/interacting partners relevant for DNA repair pathway choice

To get better insight into the interactome of REV7, we performed a standard affinity purification (AP) followed by MS (AP-MS; Fig EV1A), where REV7 was tagged with the Flag epitope and stably expressed in the human embryonic kidney 293 (HEK293) cell line using the Flp-In/T-REX system (Fig EV1B). As a complementary approach, we used a proximity-based biotin labeling technique (BioID), which allows the monitoring of proximal/transient interactions (Fig EV1A; Roux *et al*, 2012, 2013). Briefly, REV7 was fused to a mutant of an *Escherichia coli* biotin-conjugating enzyme (BirA*) and stably expressed in HEK293 as previously described (Lambert *et al*, 2015). This fusion protein is capable of biotinylating proteins that come in close proximity or directly interact with REV7 (Fig EV1C). Labeled proteins were subsequently purified by streptavidin affinity and identified by MS. Both approaches were carried out in triplicate using extracts of cells treated in the absence or the presence of the radiomimetic DNA damaging drug neocarzinostatin (NCS). We identified 140 high-confidence REV7 interactors that were either common to the four experimental conditions or found in both the AP-MS and the BioID following NCS treatment (Fig 1A and Table EV1). As expected, pathways critical for mitosis and DNA repair were enriched in our list of REV7 partners (Fig EV1D). To further refine REV7 interactors, we intersected our data with previously reported proteomic profiling of REV7 (Nelson *et al*, 1999; Chen & Fang, 2001; Weterman *et al*, 2001; Guo *et al*, 2003; Iwai *et al*, 2007; Zhang *et al*, 2007; Hong *et al*, 2009; Medendorp *et al*, 2009; Tissier *et al*, 2010; Vermeulen *et al*, 2010; Listovsky & Sale, 2013; Rolland *et al*, 2014; Huttlin *et al*, 2015, 2017). Using this methodology, we obtained 11 high-confidence REV7 interactors (Figs 1B and EV1E), including the chromosome alignment-maintaining phosphoprotein (CHAMP1), a kinetochore-microtubule attachment protein that has been recently linked to REV7 and its role during mitotic progression (Itoh *et al*, 2011), and the cell-division cycle protein 20 (CDC20), a critical activator of the anaphase-promoting complex (APC/C) that allows chromatid separation and entrance into anaphase (Chen & Fang, 2001; Listovsky & Sale, 2013; Bhat *et al*, 2015). However, whether these high-confidence REV7 interactors play any role in the DSB response remains unresolved.

To ascertain the relevance of these interactors for NHEJ, we used a well-established GFP-based reporter assay that monitors total NHEJ events (Bennardo *et al*, 2008), the EJ5-GFP assay, and targeted each candidate using small interfering RNA (siRNA) pools (Fig 1C). As positive controls, we incorporated both RIF1 and REV7, which have been previously shown to impair this assay (Chapman *et al*, 2013; Escribano-Diaz *et al*, 2013; Boersma *et al*, 2015). Out of the 11 candidates tested, downregulation of seven REV7 interactors significantly impaired the restoration of the GFP signal following

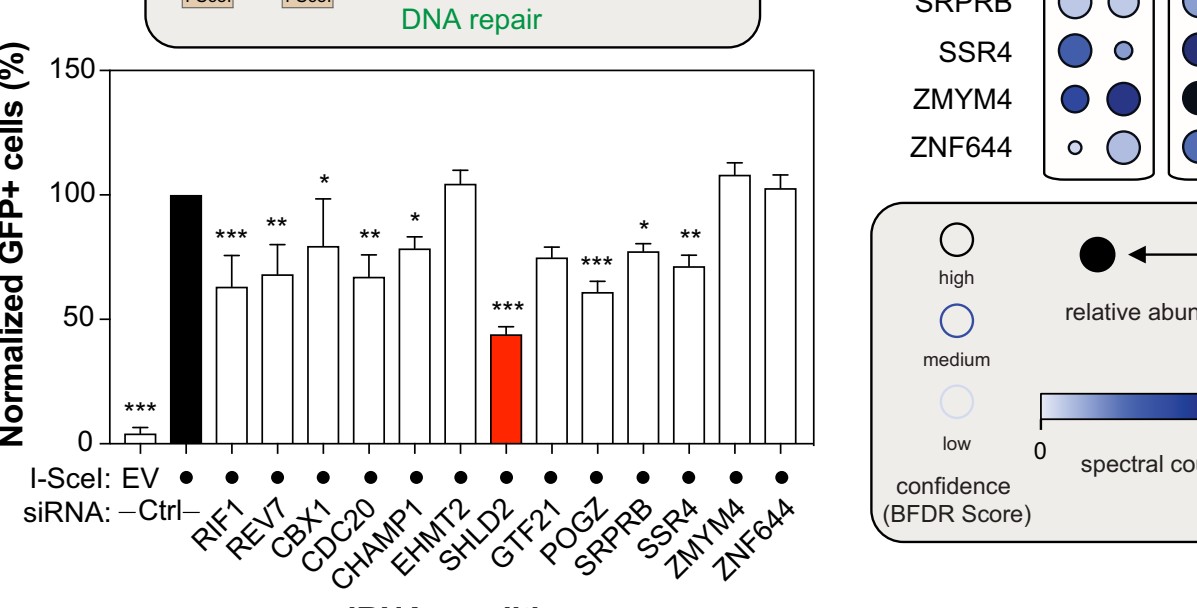

**Figure 1. Identification of novel REV7 interactors relevant for the NHEJ pathway.**

A  Venn Diagram representing the distribution of proteins identified by both the BioID and the standard AP/MS of REV7, with or without DNA damage (NCS).
B  Selected BioID REV7 results, shown as dot plots. The spectral counts for each indicated prey protein are shown as AvgSpec. Proteins were selected based on and iProphet probability of > 0.95, BFDR of < 0.05, and ≥ 10 peptide count. The circle size represents the relative abundance of preys over baits.
C  U2OS EJ5-GFP cells were transfected with the indicated siRNAs. At 24 h post-transfection, cells were transfected with the I-SceI expression plasmid, and the GFP[+] population was analyzed 48-h post-plasmid transfection. The percentage of GFP[+] cells was determined for each individual condition and subsequently normalized to the non-targeting condition (siCTRL). Data are presented as the mean ± SD (n = 3). Significance was determined by one-way ANOVA followed by a Dunnett's test. *P < 0.05, **P < 0.005, ***P < 0.0005.

DSB induction and subsequent repair by NHEJ (Fig 1C), without impacting drastically cell cycle progression (Fig EV1F). SHLD2 emerged as our strongest hit, with a reduction of more than 60% of

the GFP signal compared to the control condition in this assay (Fig 1C). Therefore, we concentrated our efforts on this factor to better define its involvement in DNA repair pathway choice.

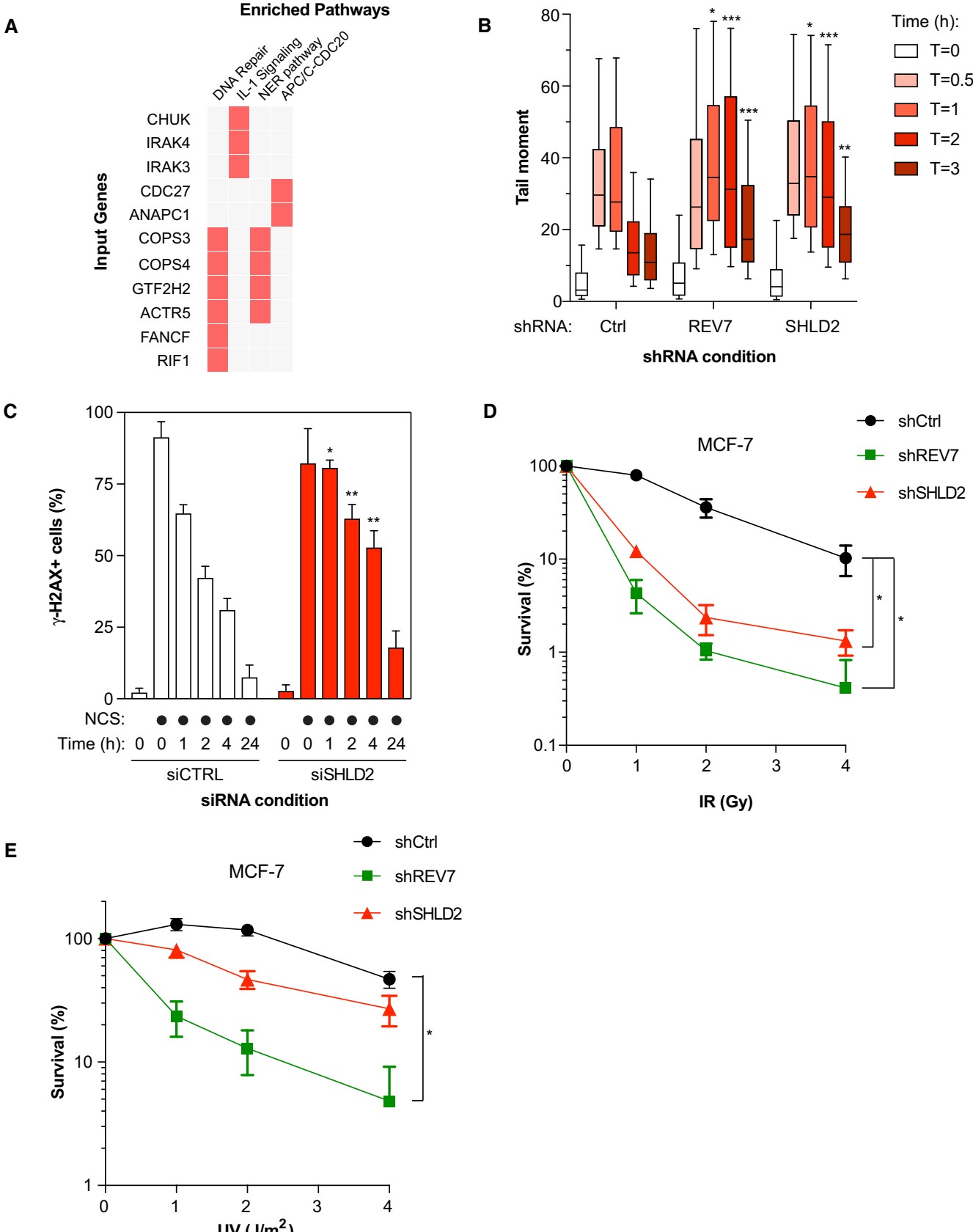

Figure 2.

**Figure 2. SHLD2 plays a critical role in DNA repair.**

A   Pathway enrichment analysis on genes co-evolving with SHLD2 in Mammalians and Vertebrates using a phylogenetic profiling approach followed by an Enrichr-based analysis.

B   Quantification of the Neutral Comet assay. U2OS cells stably expressing shCtrl, shREV7, or shSHLD2 were exposed to IR (10 Gy) and run in low melting agarose under neutral conditions. Immunofluorescence against DNA stained with SYBR Gold was performed to measure the tail moment. Data are represented as a box and whiskers graph where the box tends from the 25th to the 75th percentiles, while the whiskers are drawn down to the 10th percentile and up to the 90th. Significance was determined by one-way ANOVA followed by a Dunnett's test. *$P < 0.05$, **$P < 0.005$, ***$P < 0.0005$. For each condition, at least 50 cells were analyzed.

C   U2OS cells were transfected with the indicated siRNA; 48 h post-transfection, the cells were treated with NCS to induce DNA damage and the cells were harvested at 0-, 1-, 2-, 4-, and 24-h post-NCS treatment. Flow cytometry analysis of phosphorylated-H2AX signal was used to measure $\gamma$-H2AX endogenous signal. Data are represented as a bar graph showing the mean $\pm$ SD. Significance was determined by two-way ANOVA followed by a Bonferroni test. *$P < 0.05$, **$P < 0.005$.

D   Sensitivity to IR monitored by colony formation assay. MCF-7 cells stably expressing the indicated shRNAs were exposed to increasing doses of IR; 24 h post-irradiation, the cells were re-seeded to allow colony formation, fixed, and stained with 0.4% crystal violet. Shown is the quantification of colonies per condition which possessed more than 50 colonies. Significance was determined by one-way ANOVA followed by a Dunnett's test. *$P < 0.05$.

E   Similar to (D), except that cells were exposed to increasing doses of UV radiation. Significance was determined by one-way ANOVA followed by a Dunnett's test. *$P < 0.05$.

## SHLD2 promotes DSB repair in human cells

To get an evolutionary perspective and determine whether SHLD2 may be relevant for DNA repair, we used a novel phylogenetic profiling (PP) approach and defined the landscape of genes that co-evolved with SHLD2 among mammals and vertebrates (Tabach *et al*, 2013a). Importantly, this PP method has been previously shown to successfully predict protein function by analyzing the genes that co-evolved with a given factor of interest (Tabach *et al*, 2013a,b). Gene ontology analysis for biological process enrichment identified DNA repair, IL-1 signaling, Nucleotide Excision repair (NER), and the APC/C-CDC20 pathway as the most significant biological functions associated with genes that co-evolved with SHLD2 (Fig 2A). Strikingly, SHLD2 co-evolves with RIF1 in both mammalians and vertebrates, which further suggests a putative role of SHLD2 in the maintenance of genome stability.

To explore this hypothesis, we first assessed the ability of SHLD2 to promote DSB repair using the neutral comet assay. We depleted SHLD2 in the osteosarcoma U2OS cell line using a short hairpin RNA (shRNA; Fig EV2A), and we observed that loss of SHLD2 resulted in the persistence of comet tails (time points 1, 2, and 3 h), following exposure to ionizing radiation (IR; 10 Gy), compared to control cells (Fig 2B). For comparison, we depleted REV7 by shRNA (Fig EV2A) and obtained similar results. In a second assay, we depleted SHLD2 in U2OS cells using a deconvoluted siRNA

(Fig EV2C) and monitored the phosphorylation of the histone variant H2AX ($\gamma$-H2AX), a marker of DSBs, over time by flow cytometry following treatment with NCS. Again, the kinetics of $\gamma$-H2AX resolution was delayed in SHLD2-depleted U2OS cells compared to control conditions (Figs 2C and EV2D), suggesting a role of SHLD2 in DSB repair.

Next, we employed the breast cancer MCF-7 cell line to study the impact of SHLD2 depletion on survival following DSB induction by IR. We observed that SHLD2 depletion hypersensitizes MCF-7 cells to IR, in a manner similar to REV7 (Figs 2D and EV2A). Loss or depletion of REV7 has been previously linked to a hypersensitivity to UV in line with its TLS function (Lawrence *et al*, 1985), raising the question of whether SHLD2 is associated with a similar phenotype. Strikingly, SHLD2 depletion did not sensitize MCF-7 cells to increased doses of UV (Fig 2E), suggesting that SHLD2 is dispensable for TLS. From these results, we conclude that SHLD2 is critical for DNA repair in a TLS-independent manner.

## SHLD2 is recruited to DSBs through its N-terminal domain

SHLD2 is a 904-amino-acid protein with very limited structural information available (Fig 3A). By performing structure prediction analyses on SHLD2 protein sequence using Motif Scan (MyHits, SIB, Switzerland) and InterProScan5 (Jones *et al*, 2014), we identified a putative N-terminal DNA-binding domain (NUMOD3 domain) and a

**Figure 3. SHLD2 is recruited and accumulates at DNA damage sites.**

A   Schematic representation of SHLD2 and the different mutants used in this study. Each putative structural domain of SHLD2 is represented.

B   U2OS cells stably expressing HA-REV7 (Top) or HA-SHLD2 (Bottom) were pre-sensitized with 10 µg/ml Hoescht 33342 before exposed to UV micro-irradiation. Immunofluorescence against HA epitope and endogenous $\gamma$-H2AX was subsequently performed to monitor REV7 and SHLD2 accumulation at sites of damage. Shown are representative micrographs. Scale bar = 5 µm.

C   U2OS LacR-Fok1 cells were transfected with GFP or GFP-SHLD2, and 24 h later DNA damage was induced using Shield-1 and 4-OHT. The cells were then processed for GFP and mCherry immunofluorescence. Shown are representative micrographs. Scale bar = 5 µm.

D   Quantification of the experiments shown in (C). Data are represented as the mean $\pm$ SD ($n = 3$). At least 100 cells per condition were counted.

E   Quantification of the experiments shown in (C). Shown is the quantification of the GFP signal at the mCherry-LacR-Fok1 focus. Data are represented as a box-and-whisker plot in the style of Tukey. At least 100 cells per condition were counted. Significance was determined by non-parametric test followed by a Kruskal–Wallis test. *$P < 0.005$, **$P < 0.0005$.

F   Schematic representation of the site-directed generation of DSB by the restriction enzyme *AsiSI* (Top). 293T cell lines expressing ER-*AsiSI* with Flag-SHLD2 and treated with 1 µM of 4-OHT. 6 h later, the cells were processed and immunoprecipitated with Anti-FLAG Magnetic Beads and anti-$\gamma$-H2AX.x/Protein A/G magnetic beads. DNA was purified and subjected to qPCR detection. Shown is the quantification of IP efficiency as the percentage of DNA precipitated from input (Bottom). Data are presented as the mean $\pm$ SEM ($n = 3$). Significance was determined by two-way ANOVA followed by a Sidak test. *$P < 0.05$.

G   U2OS cells stably expressing HA-SHLD2Δ720–904 (Left) or HA-SHLD2Δ1–680 (Right) were processed as in (B). Immunofluorescence against HA epitope and endogenous RPA32 was subsequently performed to monitor RPA32 and SHLD2 accumulation at sites of damage. Shown are representative micrographs. Scale bar = 5 µm.

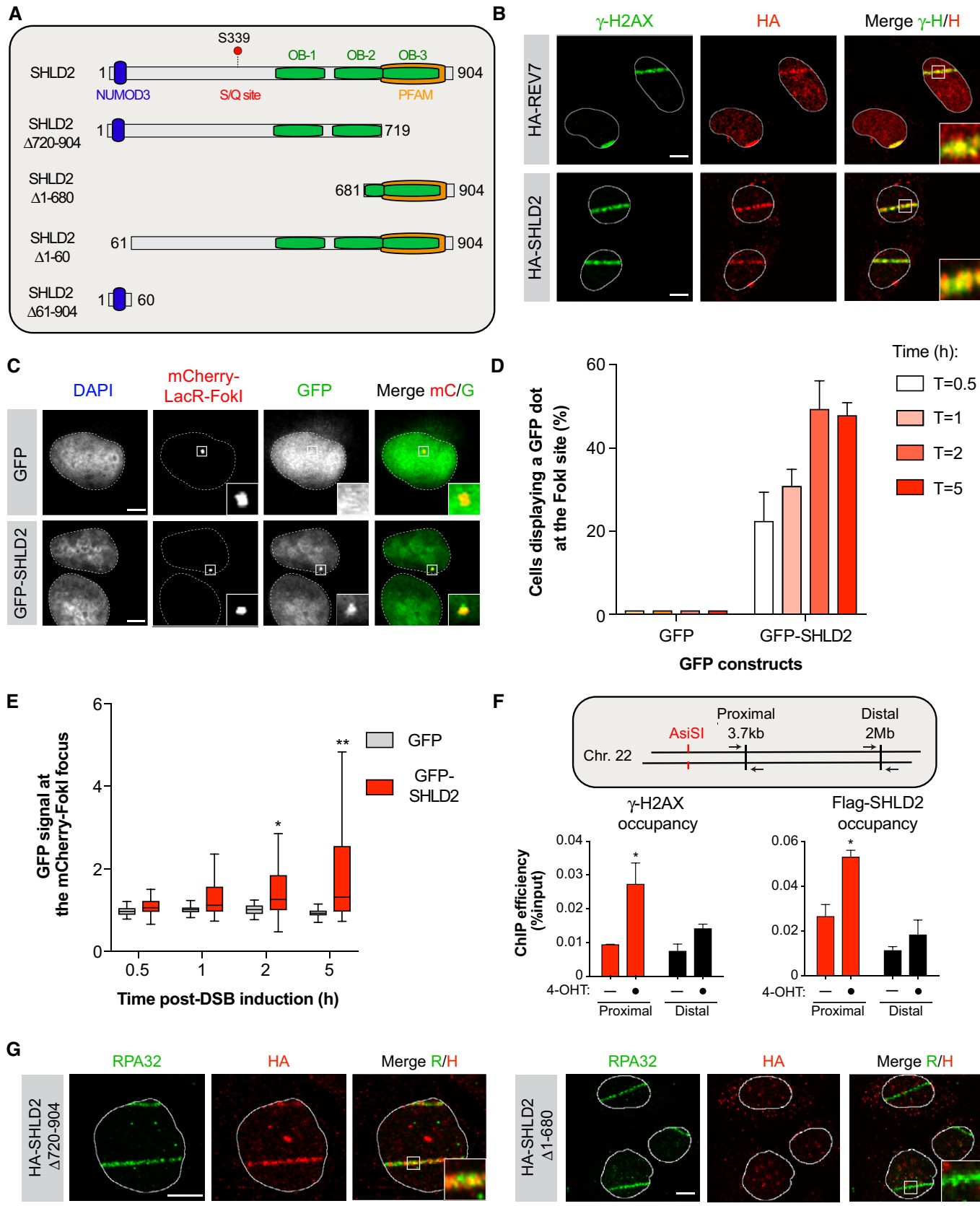

**Figure 3.**

structural motif in its C-terminus that we labeled PFAM (Fig 3A). Recently, structural prediction analyses of SHLD2 also defined a N-terminal motif promoting protein–protein as well as three putative OB-fold like domains at its C-terminus (Dev *et al*, 2018; Ghezraoui *et al*, 2018; Gupta *et al*, 2018; Noordermeer *et al*, 2018; Tomida *et al*, 2018). Finally, previous phospho-proteomic analysis identified a S/Q substrate of ATM/ATR following DNA damage at position 339 (Matsuoka *et al*, 2007).

To gain better insight into how SHLD2 promotes DSB repair, we tested whether it accumulates at DNA damage sites. Indeed, we observed that HA-tagged SHLD2, similar to REV7, is rapidly recruited to laser micro-irradiation-induced DNA damage in U2OS cells, co-localizing with γ-H2AX (Fig 3B). As laser micro-irradiation elicits high levels of both single-strand breaks (SSBs) and DSBs, we complemented this approach by using the previously described mCherry-LacR-FokI-induced DSB reporter system (Tang *et al*, 2013). Here, GFP-tagged SHLD2 is readily recruited to localized FokI-induced DSBs 30 min post-induction (Fig 3C) and the majority of cells displays a GFP-SHLD2-positive signal at the mCherry dot 2 h post-DSB induction (Fig 3D). Furthermore, GFP-SHLD2 accumulation at DSBs is significantly distinct from empty vector 2 h following the induction of DNA damage (Fig 3E). We therefore used this experimental approach for our subsequent FokI-based experiments. As an orthogonal validation of the recruitment of SHLD2 to DNA damage, we sought to use a well-established system where the induction of targeted DSBs is triggered by the controlled expression of the *Asi*SI restriction enzyme fused to a modified estrogen receptor hormone-binding domain (Iacovoni *et al*, 2010). This model has been used to monitor the recruitment of DNA repair factors in the vicinity of DSBs, and we confirmed by chromatin immunoprecipitation (ChIP) that induction of a DSB at chromosome 22 following addition of 4-hydroxytamoxifen (4-OHT) triggers the formation of γ-H2AX proximally (3.7 kb), but not distally (2 Mb) from the site of damage (Fig 3F). Importantly, we observed that Flag-tagged SHLD2 displays a similar distribution around the *Asi*SI-induced DSB. Altogether, these data indicate that SHLD2 is persistently recruited at DSBs, confirming its role in DNA repair.

We further characterized the role of SHLD2 during DSB repair by examining which domains of SHLD2 are critical for its recruitment to sites of DNA damage. We first evaluated the contribution of SHLD2 C-terminal PFAM/OB3 domain (SHLD2Δ720–904), its S/Q motif (S339), and most of SHLD2 N-terminus (SHLD2Δ1–680) for its recruitment to laser micro-irradiation-induced DNA lesions. We observed that both SHLD2Δ720–904 and S339A mutants are still recruited to DNA damage sites (Figs 3G and EV3A). However, deletion of the first 680 amino acids of SHLD2 (SHLD2Δ1–680) impairs its recruitment to laser micro-irradiation-induced DNA damage sites, suggesting a putative contribution of SHLD2 N-terminus for its accumulation at DSBs. To more quantitatively ascertain the importance of the different domains of SHLD2, we complemented this approach by monitoring the accumulation of different GFP-SHLD2 constructs in the FokI system. We observed that deletion of the first 60 amino acids of SHLD2 (SHLD2Δ1–60) totally abrogates its accumulation to DSBs, while the localization of the SHLD2 S/Q mutant (S339A) remained unaltered (Fig EV3B). Indeed, most of SHLD2 N-terminus (SHLD2Δ61–904) retained its

ability to accumulate to DSBs, suggesting a limited contribution of SHLD2 C-terminus to its recruitment at DNA damage sites. From the data, we conclude that the N-terminal domain of SHLD2 is critical for its recruitment to DSBs.

These observations suggest that SHLD2 may have the capacity to directly bind DNA. To test this hypothesis, we purified recombinant full-length SHLD2 (SHLD2-FL) from Sf9 insect cells (Fig EV3C) and monitored its capacity to bind *in vitro* single-stranded (SS) and double-stranded (DS) radiolabeled DNA probes. Interestingly, we found that SHLD2 is proficient in binding both substrates *in vitro* (Fig EV3D). Furthermore, we observed that deleting a large portion of SHLD2 C-terminus (SHLD2Δ130−904) greatly impairs its DNA-binding capacity, while the N-terminus of SHLD2 (SHLD2Δ1−129) is largely dispensable for interacting with both substrates *in vitro* (Fig EV3D). Altogether, these data suggest that SHLD2 is composed of a DSB-recruitment motif at its N-terminus and a DNA-binding domain at its C-terminus.

### SHLD2 associates with REV7 to promote NHEJ and limit HR

To decipher the link between SHLD2 and REV7, we tested the genetic requirements for the recruitment of SHLD2 to DSBs using the FokI system. Depletion of 53BP1, RIF1, or REV7 by siRNA impaired its recruitment to a localized site of DNA damage (Figs 4A and EV4A). However, we did not observe any impact on the recruitment of SHLD2 to the FokI site following BRCA1 depletion (Figs 4A and EV4A). Importantly, depletion of SHLD2 did not significantly impact the recruitment of 53BP1, RIF1, or REV7 to DSBs (Fig EV4B). These data indicate that SHLD2 is acting in concert with REV7 in the NHEJ pathway.

We reasoned that if SHLD2 is a direct effector of REV7, its recruitment to DSBs should be mediated through a physical interaction with REV7. Indeed, we confirmed the REV7-SHLD2 interaction in co-immunoprecipitation experiments where tagged versions of both REV7 and SHLD2 were expressed in 293T cells (Fig 4B). Exposure to IR did not stimulate the REV7-SHLD2 interaction and pharmacological inhibition of ATM did not abrogate it (Fig 4B), suggesting that this interaction is constitutive and stable in 293T cells, which is consistent with our MS data.

Our data point toward a role of SHLD2 in NHEJ downstream of REV7. Therefore, we confirmed that SHLD2 depletion impairs NHEJ in the EJ5-GFP assay using two distinct siRNAs (Figs 4C and EV4C). Next, we tested whether SHLD2 and REV7 act epistatically to promote NHEJ. As expected, co-depletion of REV7 with SHLD2 did not alter further the EJ5-GFP assay compared to the individual depletion (Figs 4C and EV4C). We further defined the similarities between REV7 and SHLD2 in the NHEJ pathway by testing the role of SHLD2 in CSR. REV7 depletion has been previously shown to cause a profound defect in CSR in CH12F3-2 B cells that switch from IgM to IgA following the addition of a cocktail of cytokines (IL-4/ TGF-β/anti-CD40; CIT), which induces the expression of the cytidine deaminase AID (Nakamura *et al*, 1996), and we confirmed these data (Figs 4D and EV4D; Boersma *et al*, 2015; Xu *et al*, 2015). Using two distinct shRNAs targeting SHLD2, we observed that its depletion impairs significantly CSR at both 24 and 48 h post-activation (Figs 4D and EV4D). Importantly, this phenotype is not due to a defect in AID expression (Fig EV4D) or in cell proliferation (Fig EV4E). Together, these data suggest that SHLD2 regulates CSR

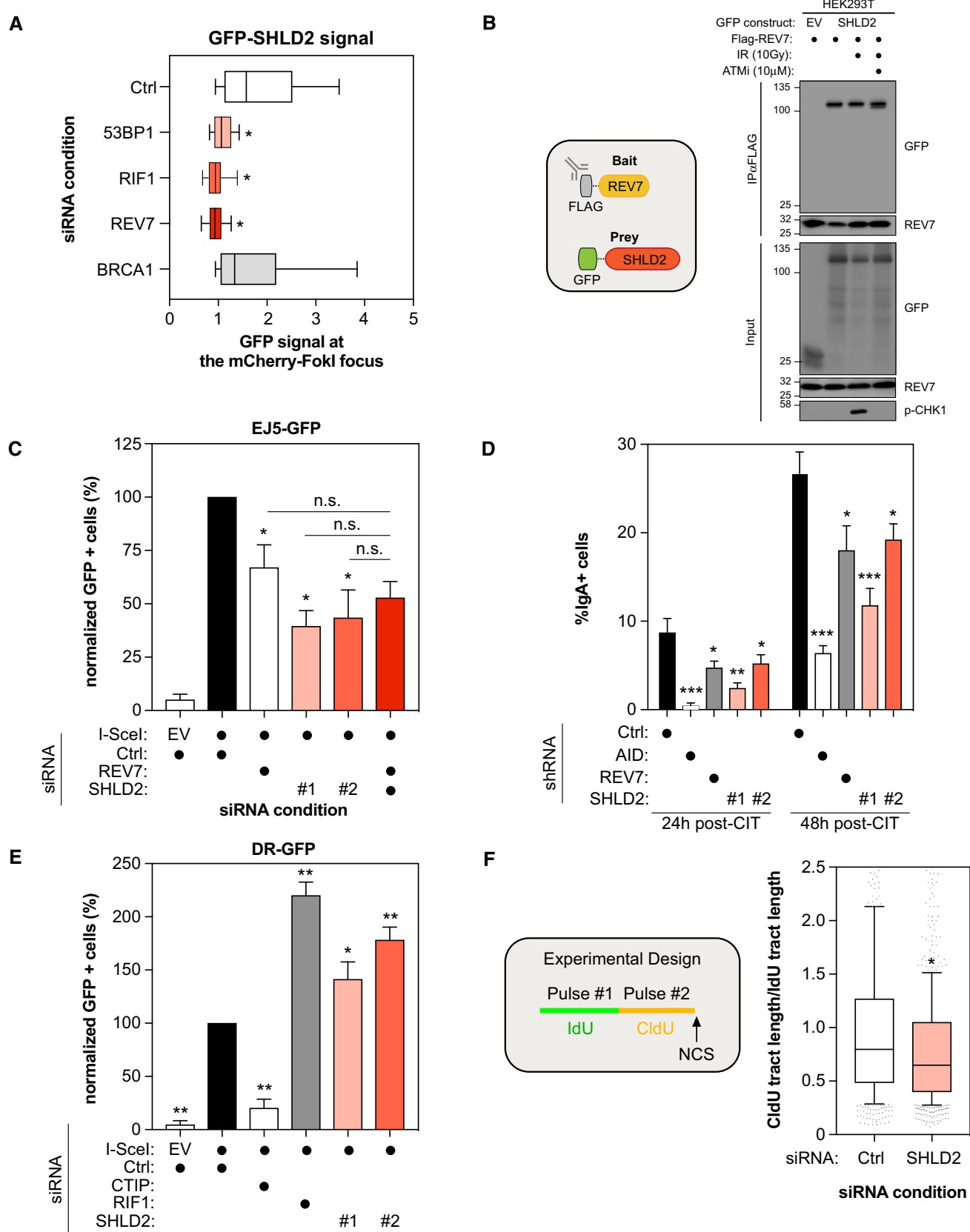

**Figure 4.**

**Figure 4.  SHLD2 is an effector of REV7 in promoting NHEJ and antagonizing HR.**

A   U2OS mCherry-LacR-Fok1 cells were treated with the indicated siRNA and subsequently transfected with a GFP-SHLD2 construct. 24 h post-transfection, DNA damage was induced using Shield-1 and 4-OHT. The cells were then fixed and analyzed for the intensity of the GFP-SHLD2 signal at mCherry-LacR-Fok1 focus. Shown is the quantification of the GFP-SHLD2 signal at the Fok1 focus. Data are represented as a box-and-whisker plot where the whiskers represent the 10–90 percentile. At least 75 cells were counted per condition. Significance was determined by one-way ANOVA followed by a Dunnett's test. $*P < 0.05$.

B   293T cells were transfected with Flag-REV7 and GFP-SHLD2 expression vectors as indicated. 24 h post-transfection cells were treated with DMSO or with 10 μM of ATM inhibitor KU-60019 for 1 h prior to irradiation. 1 h post-irradiation (10 Gy), nuclear extracts were prepared and REV7 complexes were immunoprecipitated using anti-Flag (M2) resin and then analyzed by immunoblotting using GFP, REV7, and p-Chk1 antibodies.

C   U2OS EJ5-GFP cells were transfected with the indicated siRNAs. At 24 h post-transfection, cells were transfected with the I-SceI expression plasmid, and the GFP$^+$ population was analyzed 48-h post-plasmid transfection. The percentage of GFP$^+$ cells was determined for each individual condition and subsequently normalized to the non-targeting condition (siCTRL). Data are presented as the mean ± SD ($n = 3$). Significance was determined by one-way ANOVA followed by a Dunnett's test using Ctrl+SceI as a comparison ($*P < 0.0005$) or the indicated reference (n.s. = non-significant).

D   CH12F3-2 cells stably expressing the indicated shRNAs were stimulated with a cocktail of cytokines (CIT) to induce class switching to IgA. The percentage of IgA$^+$ cells was monitored 24 and 48 h post-stimulation by staining with an anti-IgA antibody followed by flow cytometry analysis. Data are presented as the mean ± SD ($n = 3$). Significance was determined by one-way ANOVA followed by a Dunnett's test. $*P < 0.05$, $**P < 0.005$, $***P < 0.0005$.

E   U2OS DR-GFP cells were transfected with the indicated siRNAs. At 24 h post-transfection, cells were transfected with the I-SceI expression plasmid, and the GFP$^+$ population was analyzed 48-h post-plasmid transfection. The percentage of GFP$^+$ cells was determined for each individual condition and subsequently normalized to the non-targeting condition (siCTRL). Data are presented as the mean ± SD ($n = 3$). Significance was determined by one-way ANOVA followed by a Dunnett's test. $*P < 0.005$, $**P < 0.0005$.

F   Schematic representation of the DNA fiber assay experimental design (Left). U2OS cells were transfected with the indicated siRNAs and then treated with CldU, IdU, and NCS 48 h post-transfection as indicated. The slides were stained, dehydrated, mounted, and visualized and shown is the quantification of CldU/IdU tract length in order to visualize DNA end resection (Right). At least 500 DNA tracks were measured per condition. Data are represented as a box-and-whisker plot where the whiskers represent the 10–90 percentile. Significance was determined by one-way ANOVA followed by a Dunnett's test. $*P < 0.0005$.

at the level of DNA repair, which is consistent with its role as a REV7 effector.

53BP1 and its effectors have emerged as strong inhibitors of the HR pathway as well as the single-strand annealing (SSA) pathway (Chapman *et al*, 2012, 2013; Di Virgilio *et al*, 2013; Escribano-Diaz *et al*, 2013; Feng *et al*, 2013; Zimmermann *et al*, 2013; Boersma *et al*, 2015; Xu *et al*, 2015). Therefore, we tested whether depletion of SHLD2 alters both DNA repair pathways using the DR- and the SA-GFP reporter assays, respectively (Fig EV4F; Pierce *et al*, 1999; Stark *et al*, 2004). In both U2OS and HeLa DR-GFP cells (Figs 4E and EV5A), SHLD2 depletion leads to a significant increase in HR using two distinct siRNAs, similar to what we observed with RIF1. Additionally, depletion of SHLD2, like RIF1, promotes SSA (Fig EV5B; Escribano-Diaz *et al*, 2013). This anti-HR role of 53BP1 and its effectors was attributed to a putative function in limiting DNA end resection, a key step in initiating DSB repair by HR. To define whether SHLD2 controls DNA end resection, we carried out a modified version of the DNA combing assay, where a dual-pulse labeling of the replicating DNA was performed using two distinct nucleotides analogs (IdU and CldU) before addition of NCS (Fig 4F). While the length of the IdU-labeled DNA should not be altered by DNA end resection, we hypothesized that any increase in the processing of the DNA end should result in a shorter CldU-labeled DNA track and therefore a reduced ratio of CldU/IdU track length. Indeed, depletion of SHLD2 in U2OS cells resulted in a significant reduction in the CldU/IdU ratio compared to control cells (Figs 4F and EV5C), suggesting that SHLD2 limits DNA end resection. To support this hypothesis, we monitored by immunoblot the levels of phosphorylated RPA2 at position S4 and S8, which is widely used as a marker of DNA end resection, following treatment with NCS. We observed that depletion of SHLD2 in U2OS cells increased p-RPA2 levels upon NCS treatment compared to control cells (Fig EV5D), confirming that SHLD2 opposes HR by limiting DNA end resection.

Loss of REV7 in BRCA1-deficient cells has also been shown to restore partially HR (Xu *et al*, 2015). We sought to examine whether depletion of SHLD2 could result in a similar phenotype. Therefore,

we co-depleted both BRCA1 and SHLD2 in HeLa DR-GFP cells and we found a partial and significant restoration of HR in co-depleted vs. BRCA1-depleted cells (Fig EV5E). Altogether, these results are consistent with a model where SHLD2, like 53BP1, RIF1, and REV7, promotes DSB repair by NHEJ and antagonizes HR by inhibiting DNA end resection.

## SHLD2 associates with SHLD1 to promote NHEJ

It remains largely unclear how SHLD2 promote NHEJ and limit HR, similar to 53BP1, RIF1, and REV7. Therefore, we determined the interactome of SHLD2 using the BioID approach in the presence or the absence of DNA damage (+/− NCS; Fig EV6A and B, Table EV2). Using this methodology, we identified previously described SHLD2 interactors, including REV7, the RNA-binding protein HNRNPA1, and the E3 Ubiquitin ligase TRIM25 (Fig EV6B; Roy *et al*, 2014; Hein *et al*, 2015; Choudhury *et al*, 2017). Interestingly, several members of the COP9 signalosome (COPS4 and COPS6) and the Cullin-RING E3 Ubiquitin ligase family (CUL3, CUL4B, CUL5, DDB1) emerged as high-confidence proximal interactors of SHLD2. However, by comparing both REV7 and SHLD2 BioID datasets, we did not identify any common complex of relevance for DNA repair. Therefore, we sought to undertake a more systematical and unbiased approach to identify novel DNA repair factors using the CRISPR/Cas9 technology (Fig 5A). We employed the previously described TKO.v1 sgRNA library that contains 91,320 sequences and targets 17,232 genes and applied it to an hTERT immortalized retinal pigment epithelial RPE1 cell line stably expressing Cas9 (Hart *et al*, 2015). To identify genes that are relevant for DNA repair, we used the chemotherapy drug doxorubicin as a selective agent (Fig 5A). TP53, along with CHK1 and TOP2A, emerged as our strongest hits providing resistance to doxorubicin (Fig 5B, Table EV3). Their depletion was previously shown to elicit doxorubicin resistance (Burgess *et al*, 2008), thereby validating our approach (Fig 5B, Table EV3). We subsequently focused our analysis on doxorubicin sensitizers, as they are likely to play a key role in DNA repair. Interestingly, SHLD1 scored as one of

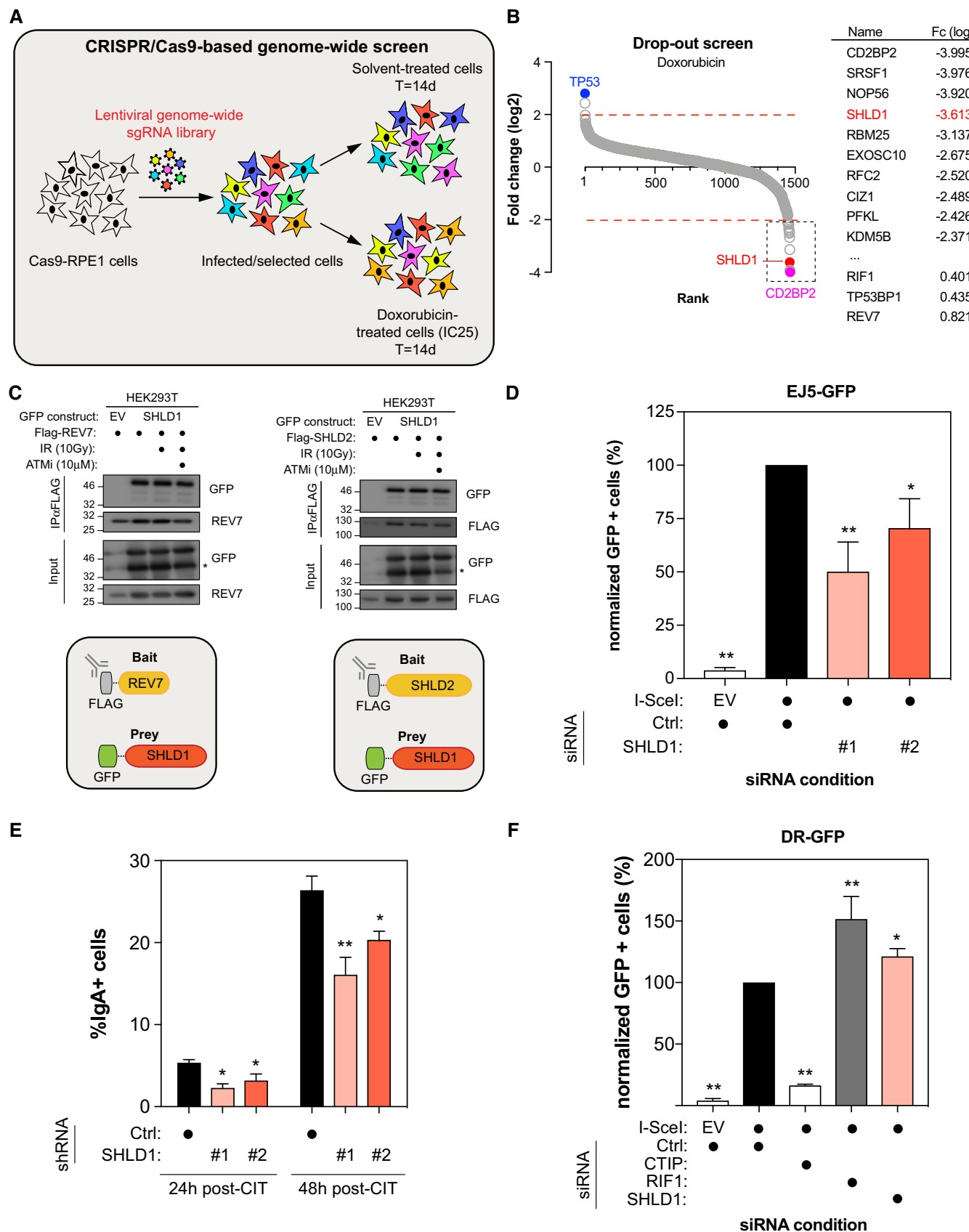

**Figure 5.**

**Figure 5.  SHLD1 co-operates with SHLD2 and REV7 to promote NHEJ and restrict HR.**

A   Schematic representation of CRISPR/Cas9-based genome-wide screen under doxorubicin treatment.

B   Genes significantly enriched or dropped out after a 14-day treatment with doxorubicin were identified by plotting as a Log2 fold change compared to untreated. Ranking was determined based on the Log2 fold score (Left). The top 10 doxorubicin sensitizers are indicated on the right with their respective fold change (Fc) in Log2.

C   293T cells were transfected with Flag-REV7 and GFP-SHLD1 (Left) or Flag-SHLD2 and GFP-SHLD1 (Right) expression vectors as indicated. 24 h post-transfection, cells were treated with DMSO or with 10 μM of ATM inhibitor KU-60019 for 1 h prior to irradiation. 1 h post-irradiation (10 Gy), nuclear extracts were prepared and REV7 or SHLD2 complexes were immunoprecipitated using anti-Flag (M2) resin and then analyzed by immunoblotting using GFP and REV7 antibodies.

D   U2OS EJ5-GFP cells were transfected with either siCTRL, siSHLD1 #1, or siSHLD1 #2. At 24 h post-transfection, cells were transfected with the I-SceI expression plasmid, and the GFP$^+$ population was analyzed 48-h post-plasmid transfection. The percentage of GFP$^+$ cells was determined for each individual condition and subsequently normalized to the non-targeting condition (siCTRL). Data are presented as the mean ± SD ($n$ = 3). Significance was determined by one-way ANOVA followed by a Dunnett's test. *$P$ < 0.005, **$P$ < 0.0005.

E   CH12F3-2 cells stably expressing either shCTRL, shSHLD1#1, or shSHLD1#2 were stimulated with a cocktail of cytokines (CIT) to induce class switching to IgA. The percentage of IgA$^+$ cells was monitored 24 and 48 h post-stimulation by staining with an anti-IgA antibody followed by flow cytometry analysis. Data are presented as the mean ± SD ($n$ = 3). Significance was determined by one-way ANOVA followed by a Dunnett's test. *$P$ < 0.05, **$P$ < 0.005.

F   U2OS DR-GFP cells were transfected with the indicated siRNAs. At 24 h post-transfection, cells were transfected with the I-SceI expression plasmid, and the GFP$^+$ population was analyzed 48 h post-plasmid transfection. The percentage of GFP$^+$ cells was determined for each individual condition and subsequently normalized to the non-targeting condition (siCTRL). Data are presented as the mean ± SD ($n$ = 3). Significance was determined by one-way ANOVA followed by a Dunnett's test. *$P$ < 0.005, **$P$ < 0.0005.

our most depleted genes (Fig 5B, Table EV3). This factor is of particular interest as it has been previously identified in the proteomic analysis of two of our high-confidence SHLD2 interactors, REV7 (Hutchins *et al*, 2010) and CUL3 (Bennett *et al*, 2010). We therefore concentrated our efforts on this factor to define its link with FAM35A during DNA repair.

First, we confirmed that REV7 and SHLD1 interact together by co-immunoprecipitation experiments where tagged versions of both REV7 and SHLD1 were expressed in 293T cells (Fig 5C). Next, we tested whether SHLD2 interacts with SHLD1 using a similar approach (Fig 5C). Importantly, both REV7-SHLD1 and SHLD2-SHLD1 interactions did not increase upon IR treatment, neither did the pharmacological inhibition of ATM abrogates them (Fig 5C), similar to what we observed previously with the REV7-SHLD2 interaction. If SHLD1 is part of a complex composed of REV7 and SHLD2, we would expect SHLD1 to accumulate at sites of damages, recapitulating the observations we made with both REV7 and SHLD2. Therefore, we carried out laser stripe micro-irradiation experiments and observed that HA-tagged SHLD1 co-localizes with γ-H2AX at DNA damages sites (Fig EV6C), suggesting a role of SHLD1 in DNA repair.

We further investigated this hypothesis by evaluating the contribution of SHLD1 in the NHEJ and in CSR. Indeed, we observed that depletion of SHLD1 in U2OS EJ5-GFP cells resulted in a significant reduction in DSB repair by NHEJ (Figs 5D and EV6D), as previously observed with SHLD2 and REV7. Furthermore, depleting SHLD1 in CH12F3-2 B cells using two distinct shRNAs impaired significantly CSR at both 24 and 48 h post-activation (Fig 5E), suggesting a potential role of SHLD1 in DNA repair during CSR. Finally, depletion of SHLD1 in the U2OS DR-GFP cells led to a significant increase in HR (Fig 5F). Altogether, these data suggest that SHLD2 functions as part of a large multi-protein complex, composed of at least REV7 and SHLD1, to promote NHEJ and CSR while restricting HR.

**SHLD2 levels correlate with a poorer prognosis in a subset of breast cancer patients**

Dysregulation of DSB repair pathways has been frequently observed in several types of cancer and extensively documented for its role in the pathobiology of breast cancer (BC). We sought to determine whether SHLD2 may contribute to the outcome of BC by interrogating two distinct patient-based cohorts of triple negative breast cancer (TNBC) and basal-like BC. Interestingly, high levels of SHLD2 correlate with a poorer survival probability in a well-annotated cohort of 24 TNBC patients (Fig 6A). We confirmed this observation in the publicly available TCGA database where we focused our analysis of basal-like BC patients. Again, high expressers of SHLD2 have significantly lower relapse-free survival in this cohort (Fig 6B), suggesting a putative role of SHLD2 in the pathobiology of a BC subset. Altogether, our data are consistent with a model where SHLD2 needs to be tightly regulated to control DNA repair pathway choice where it acts in concert with SHLD1 as a downstream effector of REV7 in the NHEJ pathway and restricts DNA end resection, thereby antagonizing HR (Fig 6C).

# Discussion

Two main DNA repair pathways, NHEJ and HR, are typically mobilized to repair cytotoxic DSBs, and optimal pathway selection is central in preserving genome integrity. Several factors, including 53BP1, RIF1, and REV7, emerged recently as key players in DNA repair pathway choice (Chapman *et al*, 2012, 2013; Di Virgilio *et al*, 2013; Escribano-Diaz *et al*, 2013; Feng *et al*, 2013; Zimmermann *et al*, 2013; Boersma *et al*, 2015; Xu *et al*, 2015). However, it remains largely unclear how they modulate the proper balance between NHEJ and HR. Several recent studies have recently tackled to decipher the effectors of the 53BP1-RIF1-REV7 axis (Barazas *et al*, 2018; Dev *et al*, 2018; Ghezraoui *et al*, 2018; Gupta *et al*, 2018; Mirman *et al*, 2018; Noordermeer *et al*, 2018; Tomida *et al*, 2018), and our work contributes to this effort by providing further insight into the role of REV7 in DNA repair pathway choice.

Using a mass spectrometry-based approach, we identified SHLD2 as a high-confidence interactor of REV7. While this association has been previously reported (Hein *et al*, 2015), it is only recently that its biological relevance has been further investigated (Dev *et al*, 2018; Ghezraoui *et al*, 2018; Gupta *et al*, 2018; Mirman *et al*, 2018; Noordermeer *et al*, 2018; Tomida *et al*, 2018). Up to now, SHLD2 remained an enigma in regard to its physiological functions. The first indication of a potential involvement of SHLD2 in the response to DSBs emerged from a comprehensive interactome mapping of

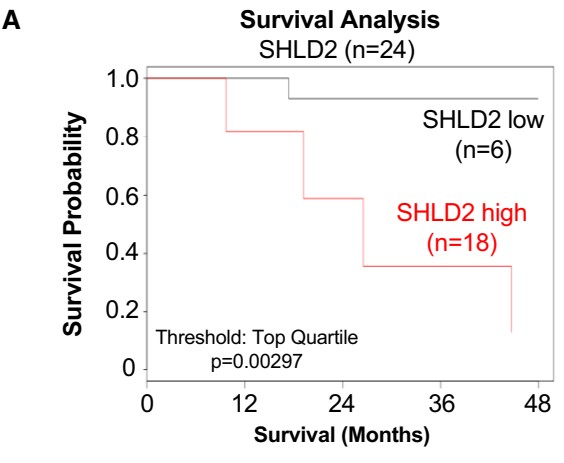

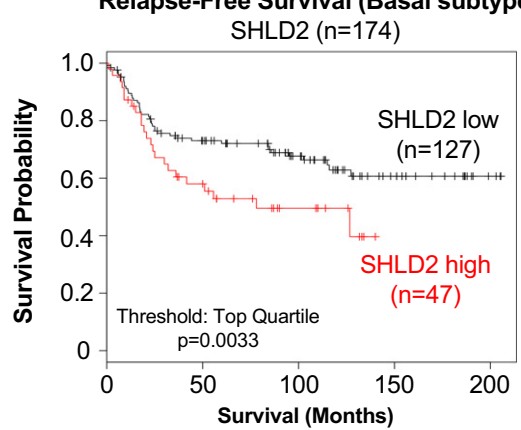

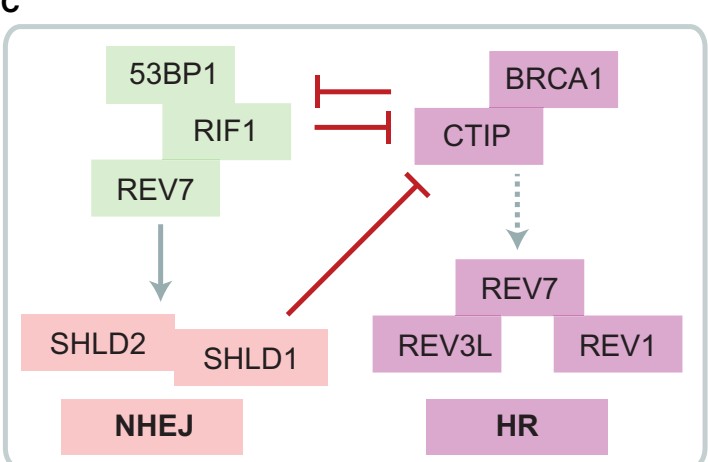

**Figure 6. SHLD2 levels are candidate marker for the prognosis of subset of breast cancer.**

A  Survival analysis of low and high expressers of SHLD2 in a cohort of 24 patients affected by triple negative breast cancer (TNBC). Data are represented as Kaplan–Meyer curves with expression classified as low and high. Threshold for high cutoff is the top quartile. Significance was determined by calculating the hazard ratio with 95% confidence and the logrank P-value.

B  Relapse-free survival (Basal Subtype) of low and high expressers of SHLD2 obtained from the KM-plotter database. Data are represented as Kaplan–Meyer curves with expression classified as low and high. Threshold for high cutoff is the top quartile. Significance was determined by calculating the hazard ratio with 95% confidence and the logrank P-value.

C  Schematic incorporating SHLD2 and SHLD1 as REV7 effectors in the NHEJ pathway and modulators of DNA repair pathway choice.

key DNA repair factors, including 53BP1 and BRCA1 (Gupta *et al*, 2018). Here, we provide further insight into the role of SHLD2 in DNA repair and show that SHLD2 acts as a downstream effector of REV7 in the NHEJ pathway. Through its N-terminal domain, SHLD2 is mobilized to and accumulates at sites of DNA damage in a 53BP1-, RIF1-, and REV7-dependent manner, in accordance with several recent studies describing the role of SHLD2 in the DNA damage response (Barazas *et al*, 2018; Dev *et al*, 2018; Ghezraoui *et al*, 2018; Gupta *et al*, 2018; Mirman *et al*, 2018; Noordermeer *et al*, 2018; Tomida *et al*, 2018). Importantly, we show that the N-terminus of SHLD2 has very limited DNA-binding capacity, which support a model where the recruitment of SHDL2 to DSBs is promoted by protein–protein interactions (Noordermeer *et al*, 2018). This finding corroborates the initial observation made by Gupta *et al* (2018), showing that the N-terminal domain of SHLD2 is

critical for its association with REV7. In a series of functional studies, we show that SHLD2 is critical during both antibody diversification and DSB repair by the NHEJ pathway. Our data suggest that SHLD2 and REV7 act together in an epistatic manner, which is corroborated by several studies that described SHLD2 as a novel DNA repair factor (Barazas *et al*, 2018; Dev *et al*, 2018; Ghezraoui *et al*, 2018; Gupta *et al*, 2018; Mirman *et al*, 2018; Noordermeer *et al*, 2018; Tomida *et al*, 2018). We further show that, similar to 53BP1 and RIF1 (Chapman *et al*, 2012, 2013; Di Virgilio *et al*, 2013; Escribano-Diaz *et al*, 2013; Feng *et al*, 2013; Zimmermann *et al*, 2013), SHLD2 opposes HR by limiting DNA end resection. However, whether this anti-HR function of SHLD2 is related to a steric hindrance of the DNA ends or an active process of preventing CtIP and its associated nucleases to initiate DNA end resection remains an avenue of investigation.

Second, our genome-wide screening approach identified novel players in DNA repair, including SHLD1. Our data point toward a more complex model where REV7, SHLD2, and SHLD1 cooperate together to promote NHEJ and limit HR, as described by several other groups (Barazas *et al*, 2018; Dev *et al*, 2018; Ghezraoui *et al*, 2018; Gupta *et al*, 2018; Mirman *et al*, 2018; Noordermeer *et al*, 2018; Tomida *et al*, 2018). Indeed, we show that SHLD1 co-immunoprecipitates with both REV7 (as previously described (Hutchins *et al*, 2010)) but also SHLD2. Surprisingly though, we did not identify SHLD1 as a high-confidence interactor of SHLD2 in our BioID approach, likely due to the low abundance of this factor and in accordance with a previous report (Gupta *et al*, 2018). Still, our genetic dissection of SHLD1 recapitulates the striking data that we observe with SHLD2: (i) SHLD1 is recruited to and accumulates at sites of DNA damage; (ii) its depletion impairs both NHEJ and CSR while promoting HR. Why REV7 requires several factors to promote NHEJ and inhibit DNA end resection is unclear; our model suggests that, alike the Shelterin complex at telomeres (Schmutz & de Lange, 2016), which lacks any catalytic activity *per se*, REV7 forms a large multi-protein complex at DSBs to protect DNA ends from extensive processing and promote their rapid joining by the NHEJ machinery. This model, described elegantly in Noordermeer *et al* (2018), has driven the nomenclatural renaming of the SHLD proteins as the Shieldin complex.

Finally, our observation that SHLD2 levels correlate with a poor prognosis in a subset of BC has profound implications for the diagnosis and treatment of these patients. Imbalance in DSB repair pathways has been well documented to predispose and promote the development of BC; in the majority of the cases, inactivation of HR factors is the cause of this predisposition with a very limited understanding of the molecular mechanisms underlining this phenomenon. Our study points toward an expressional dysregulation of SHLD2 as a potential predisposing factor to TNBC/Basal-like BC outcome, which may point toward a direct contribution of this novel NHEJ component in the pathobiology of BC. It will be of great importance to further define the role of SHLD2 in BC as it may be a relevant biomarker for its diagnosis.

Altogether, the work presented here not only describes the role of two DNA repair factors in controlling DNA repair pathway choice, but it also provides the first evidence that SHLD2 could benefit clinicians as a relevant biomarker for a subset of BC. Our data point toward a more complex model for DNA repair pathway choice where REV7 mobilizes additional factors to DSBs to catalyze NHEJ and limit the processing of DNA ends, thereby restricting HR to the S/G2 phases of the cell cycle.

# Materials and Methods

### Cell culture and plasmid transfection

HEK293-T, Flp-In T-REx, -XT, and HeLa cells were cultured in Dulbecco's modified Eagle's medium (DMEM; Wisent) and were supplemented with 10% fetal bovine serum (FBS) and 1% penicillin–streptavidin (P/S). CH12F3-2 cells were cultured in RPMI 1640 (Wisent) supplemented with 10% FBS, 5% NCTC-109 media (Thermo Fisher), 50 μM 2-mercaptoethanol, and 1% P/S. U2OS cells were cultured in McCoy's 5A modified medium (Wisent) and was supplemented with 10% FSB and 1% P/S. All cell lines were tested for mycoplasma contamination and STR DNA authenticated. Plasmid transfections were carried out using Lipofectamine 2000 Transfection Reagent (Invitrogen) following the manufacturer's protocol. Lentiviral infections were done as previously described (Escribano-Diaz *et al*, 2013), with modifications listed below.

To generate the ER-AsiSI-expressing HEK293T cell line, retroviral particles were produced using pBABE HA-AsiSI-ER (kind gift of Dr. Michael Witcher, McGill University), the packaging plasmid pUMVC (addgene 8449), and the envelope plasmid VSV-g (Addgene #8454) co-transfected into HEK293T cells. To produce U2OS stable cell lines for micro-irradiation, lentiviral particles were produced by polyethyleneimine-mediated transfection of pHAGE-EF1α plasmid-cDNA with psPAX2 and pMD2.G packaging and envelope plasmids in 293 XT packaging cells. U2OS cells were infected with lentiviruses for 24 h in media containing 8 μg/ml polybrene, and 1 μg/ml puromycin selection was applied. To generate a HEK293-TREx, Flp-in cells were co-transfected with pOG44 and pcDNA5 FRT/TO-FLAG-FAM35A and selected in with 200 μg/ml hygromycin B and 5 μg/ml blasticidin. The U2OS-LacI-FokI-mCherry cell line was a kind gift of R. Greenberg (University of Pennsylvania). The DNA repair reporter cell lines DR-GFP, EJ5-GFP, and SA-GFP were a kind gift of Dr. Jeremy Stark (City of Hope National Medical Center, California).

### Plasmids

The cDNAs of human SHLD2, SHLD1, and REV7 were obtained from Sidong Huang (McGill University). Quikchange site-directed mutagenesis (Agilent) was performed as per manufacturer's guidelines to obtain the different SHLD2 mutants. All these constructs were transferred from ENTRY vectors into lentiviral pHAGE-EF1α vectors in frame with N-terminal 3XHA epitopes, GFP-construct, BirA-construct, and Flag-tagged constructs using LR Clonase II according to manufacturer's instructions (ThermoFisher). Plasmids encoding, I-SceI or pDEST-FRT-FLAG (EV) for the different GFP reporter assays, were kindly provided by Dr. Daniel Durocher (Lunenfeld-Tanenbaum Research Institute). The following pLKO-puro shRNA lentiviruses obtained from Mission library clones (Sigma) against mouse genes: Negative control (scramble); *Mad2 l2* 45 (TRCN0000012845); *Aicda* (TRCN00000112033); *Fam35a (Shld2)* (TRCN0000183111) and *Fam35a (Shld2) (*TRCN0000183379); *C20orf196* (*Shld1*) (TRCN0000092993) and *C20orf196* (*Shld1*) (TRCN0000092995).

### RNA interference

All siRNAs employed in this study were single duplex siRNAs purchased from Dharmacon (GE Healthcare, Colorado, USA). RNAi transfections were performed using Lipofectamine RNAiMax (Invitrogen) in a forward transfection mode. Except when stated otherwise, siRNAs were transfected 48 h prior to cell processing. The individual siRNA duplexes used are as follows: Control (D-001810-03); RIF1 (D-027983-02); CBX1 (L-009716-00); CDC20 (L-021601-02); CHAMP1 (L-021601-02); EHMT2 (L-006937-00); GTF21 (L-013686-00); POGZ (J-006953-10/12); REV1 (D-008234-01/02/0/3/04); REV3L (D-006302-01/02/03/04); REV7 (J-003272-14); SRPRB (L-013646-00); SSR4 (L-012264-00); ZMFM4 (L-019932-02); ZNF644 (L-007085-02); SHLD2 (D-013761-01/02/03/04); SHLD1 (D-018767-01/02/03/04); BRCA1 (D-003461-05); CTIP (M-011376-00). In most

of the experiments, SHLD2 siRNAs D-013761-01 and D-013761-03 were used except during the validation screen.

## Immunofluorescence microscopy

In most cases, cells were grown on glass coverslips. All steps were carried out at room temperature. Cells grown on coverslips were fixed in freshly prepared 2% paraformaldehyde for 10 min. Fixed cells were then incubated for 10 min with a combination of permeabilization/blocking buffer [0.1% Triton X-100 and 1% bovine serum albumin (BSA)]. Next, primary antibodies were added for 1.5 h in phosphate-buffered saline (PBS) + 1% BSA followed by three washes with PBS. Secondary antibody was next added in the same buffer for a period of 1 h. Nuclei were stained with DAPI (1 μg/ml) for 5 min and subjected to a set of final washes with PBS and subsequently sterile water. After this, coverslips were mounted onto glass slides using a ProLong Diamond antifade reagent (Life Technologies). Images were acquired using a Zeiss LSM800 confocal microscope. Images were analyzed and quantified using ImageJ software [National Institutes of Health (NIH)]. For the FokI system, DSBs were induced by adding Shield-1 and 4-OHT for 2 h prior to immunofluorescence sample preparation.

## Clonogenic assay

Clonogenic assays were performed as described (Orthwein *et al*, 2014). Briefly, cells were allowed to reach ~50% confluence prior to genotoxic insult. Culture plates were then exposed to the indicated dose of IR or 254 nm ultraviolet (UV) light and allowed to recover overnight. Cells were then trypsinized and re-seeded into 60-cm dishes at 400 cells (or 800 cells for the highest dose) per dish. Colonies were allowed to form over the duration of 2 weeks and then fixed in 100% methanol and stained with 0.4% crystal violet (in 20% methanol). Colony number was manually tabulated with only colonies of > 50 cells included in the total count.

## Co-immunoprecipitation

HEK293T cells were co-transfected with pDEST-FRT/TO-FLAG and pDEST-FRT-TO-GFP-tagged vectors. Twenty-four hour post-transfection, all cells were exposed to irradiation (10 Gy). ATMi-treated cells were exposed to 10 μM KU-60019 1 h prior to irradiation. Cells recovered at 37°C for 1 h before being harvested and lysed in a high salt lysis buffer (50 mM Tris, 300 mM NaCl, 1 mM EDTA, 1% Triton X-100), supplemented with 1× Protease Inhibitor Cocktail (Roche)/Phosphate Inhibitor cocktail (Sigma) and gently rotated for 30 min at 4°C. Nuclear fractions were extracted with 0.25 M CaCl$_2$ and 250 U benzonase and homogenized on an orbital shaker for 15 min at 30°C. The resulting solution was pelleted at 4°C at 18,000 *g* for 15 min, and the supernatant was applied to an anti-Flag (M2) resin (Sigma) and equilibrated at 4°C for 2 h. The anti-Flag resin was then washed once with the high salt lysis buffer and twice with the immunoprecipitation (IP) buffer (50 mM Tris, pH 7.6, 150 mM NaCl, 1 mM EDTA). The immunoprecipitated proteins were eluted from the resin with 1× LDS NuPage sample buffer (10 mM Tris–HCl, 140 mM Tris-base, 0.5 mM EDTA, 1% lithium dodecyl sulfate, 10% glycerol).

## Biotin labeling and sample preparation for MS

Samples for BioID were processed as previously described (Lambert *et al*, 2015). Briefly, HEK293-T cells were either transiently transfected with FLAG-BirA*-SHLD2 or stably expressed using the T-REX system (FLAG-BirA*-REV7). Media was supplemented, 24 h post-transfection, with 50 μM biotin, and cells were incubated for an additional 24 h with neocarzinostatin (NCS, 150 ng/ml). Cells were then harvested, washed twice with PBS, and dried. Pellets were subsequently resuspended in cold RIPA buffer containing: 50 mM Tris–HCl pH 7.4, 150 mM NaCl, 1 mM EDTA, 1% NP-40, 0.1% SDS, 0.5% sodium deoxycholate, 1 mM PMSF, 1 mM dithiothreitol, 1:500 Sigma-Aldrich protease inhibitor cocktail P8340. Cell homogenates were sonicated, followed by the addition of 250 U benzonase, and centrifuged (12,000 *g*, 30 min). Supernatants were incubated with pre-washed streptavidin-sepharose beads (GE, #17-5113-01) at 4°C with rotation for 3 h. Beads were collected by centrifugation (425 *g*, 1 min), washed twice with RIPA buffer, three times with 50 mM ammonium bicarbonate (ABC, pH 8.2). Beads were resuspended in 50 mM ABC and treated with 1 μg trypsin (Sigma-Aldrich T6567) overnight at 37°C with rotation. Digestion was continued by adding an additional 1 μg of trypsin for an additional 2 h at 37°C with rotation. Supernatant containing peptides, and supernatants from two following washes with HPLC-grade H$_2$O, was collected and pooled. Digestion was ended with the addition of formic acid to a final concentration of 5%. Samples were centrifuged (18,500 *g* for 10 min), and the supernatants were dried in a SpeedVac for 3 h at high rate. Peptides were resuspended in 5% formic acid and kept at −80°C for mass spectrometric analysis. MS processing and protein analysis were carried out as previously described.

Mass spectrometry data generated by the Regional Mass Spectrometry Centre (Université de Montréal) or the IRCM Proteomics Discovery Platform were stored, accessed, searched, and analyzed using the ProHits laboratory information management system (LIMS) platform. Significance Analysis of INTeractome (SAINT) express (v3.6.1) was the statistical tool utilized to calculate the probability of protein–protein interaction from background, non-specific interactions (Choi *et al*, 2011). These results were evaluated with the Trans-Proteomic Pipeline (TPP v5.1) via the iProphet search engine integrated in the ProHits software (Liu *et al*, 2010; Shteynberg *et al*, 2011). A minimum of two unique peptide ions, an iProphet probability of > 0.95, a bait false discovery rate (BFDR) of < 0.05, and a ≥ 10 peptide count were the criteria required for protein consideration. Biofilters were applied against albumin, artifact protein, cytoskeleton, and keratin. Resultant proteins from AP-MS (FLAG-REV7) and BioID (BirA-REV7; BirA-SHLD2) experiments were tabulated and analyzed for common potential interactors between the AP-MS and BioID groups, respectively. Common candidates were then sorted according to largest peptide counts. This analysis yielded 140 mutual candidates for the NCS$^+$ group and 170 for the NCS$^−$ group. By way of literature review and the use of the Biological General Repository for Interaction Datasets (BioGRID), promising candidates were selected for targeted experimentation. Selected prey proteins were used for dot-plot heat map generation. Plots were generated using the ProHits Visualization Suite (ProHits-viz; Knight *et al*, 2017).

## GFP-based DNA repair assays

For DR-, EJ5-, EJ2-, SA-GFP reporter assays, U2OS or HeLa cells carrying the respective GFP expression cassette were transfected with indicated siRNA. Twenty-four hours after transfection, cells were transfected with empty vector (EV, pDEST-FRT-FLAG) or I-SceI plasmids. After 48 h, cells were trypsinized, harvested, washed, and resuspended in PBS. The percentage of GFP-positive cells was determined by flow cytometry. The data were analyzed using the FlowJo software.

## Class switch recombination assay

Immunoglobulin (Ig) M (IgM) to IgA switching was assayed in CH12F3-2 cells with integrated shRNA for REV7, AID, SHLD2, or SHLD1. Cells were activated in 1 ml complete CH12F3-2 media with 1.25 ng transforming growth factor beta 1 (TGF-β1, PeproTech), 5 ng interleukin (IL) 4 (IL-4, PeproTech), and 0.5 μg anti-cluster of differentiation (CD) 40 (CD40, eBioscience). IgA expression was measured by flow cytometry using primary conjugated anti-mouse IgA-PE (Southern Biotech) at 24 and 48 h after activation. Proliferation of the different transduced CH12F3-2 cell lines was monitored using carboxyfluorescein succinimidyl ester (CFSE, Invitrogen) following the manufacturer's guidelines. Class switching assays were done in triplicate for every independent experiment.

## Comet assay

U2OS cells were exposed to IR (10 Gy) and processed according to manufacturer's recommendations (Trevigen). Cells were trypsinized at the indicated time points and resuspended at $10^5$ cells/ml in PBS. Cells were combined with low melting agarose at 1:10 ratio and spread over the CometSlide. Slides were allowed to dry at 4°C for 10 min, then immersed in lysis buffer (Trevigen) overnight. The next day, the slides were immersed neutral electrophoresis buffer (two 15-min washes) followed by electrophoresis at 31 V for 45 min. Subsequently, the slides were incubated for 30 min in DNA precipitation solution followed by 30 min in 70% ethanol. Slides were dried and stained with SYBR Gold (Invitrogen). Images were taken using the EVOS FL Cell Imaging System microscope, and the tail moment was quantified using the CaspLab software. For each condition, at least 50 cells were analyzed.

## Immunoblot

Cells were washed with cold PBS (2×), and whole cell lysates were collected using: 50 mM HEPES, KOH (pH 8.0), 100 mM KCl, 2 mM EDTA, 0.1% NP-40, 10% glycerol, and protease/phosphatase inhibitors (Tkac *et al*, 2016). The following antibodies were used for the immunoblot analysis: rabbit anti-pRPA2 S4/S8 (A300-245A, Bethyl), and mouse anti-α-tubulin (ab7291, Abcam).

## Laser micro-irradiation

U2OS stable cell populations expressing the various constructs were transferred to a 96-well plate with 170 μm glass bottom (Ibidi), pre-sensitized with 10 μg/ml Hoescht 33342, and microirradiated using a FV-3000 Olympus confocal microscope equipped with a 405 nm laser line as described previously (Gaudreau-Lapierre *et al*, 2018). Immunofluorescence was performed as described previously (Gaudreau-Lapierre *et al*, 2018). Briefly, following micro-irradiation, cells were allowed to recover before pre-extraction in 1× PBS containing 0.5% Triton X-100 on ice for 5 min. Following washes with 1× PBS, cells were fixed for 15 min in 3% paraformaldehyde 2% sucrose 1× PBS solution, permeabilized in 1× PBS containing 0.5% Triton X-100 for 5 min, blocked in 1× PBS containing 3% BSA and 0.05% Tween-20, and stained with the following primary antibodies 1:500 RPA32 mouse (Santa Cruz, sc-56770) or 1:500 γ-H2A.X mouse (abcam, ab26350) and 1:500 HA-tag rabbit (Bethyl, A190-108A). After extensive washing, samples were incubated with 1:250 each of goat anti-mouse Alexa 488-conjugated and goat anti-rabbit Alexa 647-conjugated antibodies (Cell Signaling 4408S and 4414S). DAPI staining was performed, and samples were imaged on a FV-3000 Olympus confocal microscope.

## ChIP quantitative PCR

Stable 293T cell lines expressing ER-AsiSI cells were transfected with Flag-SHLD2 and treated with 1 μM of 4-OHT (4-hydroxytamoxifen) for 6 h. Cells were collected for ChIP assay as per previously (Iacovoni *et al*, 2010). Briefly, cells were crosslinked using 1.5 mM EGS [ethylene glycol bis(succinimidyl succinate), Thermo Fisher # 21565], followed by 1% of formaldehyde. Cell nuclei were isolated and lysed. Chromatin was sonicated for 15 min using a water bath sonicator/bioruptor. Fragmented chromatin bound to SHLD2 and γ-H2AX was immunoprecipitated using Anti-FLAG Magnetic Beads (Sigma, M8823), and anti-γ-H2AX (JBW301, EMD-Millipore, Massachusetts, USA) in combination with protein A/G magnetic Beads, respectively. Antibody/protein/DNA complexes were then eluted and reverse crosslinked. DNA was purified using QIAquick Kit (Qiagen #28106) and used for qPCR detection with the following oligonucleotides: AsiSI22-distF 5′-CCCATCTCAACCTCCA CACT-3′; AsiSI22-distR 5′-CTTGTCCAGATTCGCTGTGA-3′; AsiSI22-ProxF 5′-CCTTCTTTCCCAGTGGTTCA-3′; AsiSI22-ProxR 5′-GTGG TCTGACCCAGAGTGGT-3′. IP efficiency was calculated as percentage of input DNA immunoprecipitated.

## DNA fiber combing

U2OS cells were transfected with the indicated siRNA in a 6-well cell culture plate. After 48 h, cells were treated with indicated schedules and concentrations of thymidine analogue pulses [chlorodeoxyuridine (CldU; C6891); iododeoxyuridine (IdU; I7125); Sigma, Missouri, USA] with and without neocarzinostatin (NCS, Sigma) treatment to measure replication fork kinetics and extent of DNA end resection. Cells were trypsinized, agarose plug embedded, and subjected to DNA extraction as per the Fibreprep protocol (Genomic Vision, Bagneux, FR). Vinylsilane-coated coverslips (Genomic Vision) were combed through prepared DNA solution using Fibre-Comb Molecular Combing system (Genomic Vision). Combed DNA was dehydrated, denatured, blocked with BlockAid blocking solution (Invitrogen, California, USA), and stained with mouse anti-BrdU (B44, BD, New Jersey, USA), rat anti-BrdU (BU1/75, Abcam, Cambridge, UK), and rabbit anti-ssDNA (18731, Immuno-Biological Materials, Gunma, Japan) antibodies. Slides were subsequently washed and stained with secondary antibodies: Goat anti-rabbit IgG

conjugated Alexa Fluor 480 (BD Horizon), goat anti-mouse IgG conjugated Alexa Fluor 555 (Invitrogen), goat anti-rat conjugated Cy5 (Abcam). Slides were dehydrated, mounted, and visualized using FibreScan services (Genomic Vision).

## Phospho-H2AX flow cytometry

U2OS cells were transfected with indicated siRNA in a 6-well cell culture plate. After 48 h, cells were treated with NCS for 30 min, and after indicated time intervals, cells were trypsinized, washed, and fixed with 1% paraformaldehyde, washed, and subsequently permeabilized in 70% ethanol at $-20°C$. Cells were washed twice with intracellular wash buffer (1% BSA, 0.05% Tween-20, PBS) and resuspended in 1.0 µg/ml mouse anti-γ-H2AX (JBW301, EMD-Millipore, Massachusetts, USA) for 1 h at RT. Cells were then washed and resuspended in 2.0 µg/ml goat anti-mouse Alexa Fluor 647 (Invitrogen) for 1 h at RT. Cells were washed and resuspended in a propidium iodide (PI) solution (20 µg/ml PI, 300 µg/ml RNase, PBS), incubated at RT for 30 min. Events were acquired on a LSRFortessa (BD). Events were analyzed on FlowJo v10 (Treestar, Oregon, USA).

## CRISPR/Cas9 genome-wide screen

For the genome-wide CRISPR/Cas9-based screen, 270 million RPE-hTERT/Cas9 cells were transduced as described previously (Hart *et al*, 2015) with TKOv1 concentrated library virus at MOI = 0.2, ensuring a coverage of at least 600-fold for each individual sgRNA represented in the cell population. Two days later, puromycin was added to the media at a final concentration of 15 µg/ml and incubated for 4 days to allow for the emergence of resistant cells with fully repaired sgRNA library targeted loci. Cells were then split into two pools each in triplicate at a cell density of 54 million cells/replicate and treated with either vehicle (H₂O) or doxorubicin at its IC25 (3 nM) and cultured for 2 weeks with puromycin at a concentration of 7.5 µg/ml. Cells were passaged every 3 days keeping a minimum cell concentration of 54 million cells per replicate to ensure that a 600-fold library coverage was maintained over the duration of selection. At each time point, cell pellets were collected and frozen prior to genomic DNA extraction. Cell pellets were resuspended in 6 ml DNA lysis buffer (10 mM Tris–Cl, 10 mM EDTA, 0.5% SDS, pH 8.0) with 100 µg/ml RNase A, followed by incubation at 37°C for 60 min. Proteinase K was subsequently added (400 µg/ml final), and lysates were further incubated at 55°C for 2 h. Samples were then briefly homogenized by passing them three times through a 18G needle followed by three times through a 22G needle. Sheared samples were transferred into pre-spun 15 ml MaXtract tubes (Qiagen) mixed with an equal volume of neutral phenol:chloroform:isoamyl alcohol (25:24:1) solution, shook, and centrifuged at 1,500 *g* for 5 min at RT. The aqueous phase was extracted and precipitated with two volumes of ethanol and 0.2 M NaCl. Air-dried pellets were resuspended in water and quantitated via UV absorbance spectrometry.

For next-generation sequencing (NGS), sgRNA integrated loci were amplified from 330 µg of total genomic DNA per replicate using two rounds of nested PCR. The initial outer PCR consisted of 25 cycles with an annealing temperature of 65°C using Hot start Q5 polymerase (NEB) using primers Outer Primer Forward (AGGGCCTATTTCCCATGATTCCTT) and Outer Primer Reverse (TCAAAAAAGCACCGACTCGG). PCR products were pooled, and ~2% of the input was amplified a further 12 cycles for the addition of Illumina HiSeq adapter sequences. The resulting ~200-bp product from each pooled sample was further purified following separation in a 6% 0.5× TBE polyacrylamide gel. The amplicon library NGS-ready final product was quantified using qPCR and submitted for deep-sequencing on the HiSeq 2500 Illumina platform using standard Single-Read (SR) 50-cycle chemistry with dual-indexing with Rapid Run reagents. The first 20 cycles of sequencing were "dark cycles", or base additions without imaging. The actual 26-bp read begins after the dark cycles and contains two index reads, reading the i7 first, followed by i5 sequences. Prior to analysis, FastQ NGS read files were initially processed using FastQC software to assess uniformity and quality. Reads were trimmed of NGS adapter sequences using the Cutadapt tool. Reads were aligned to the sgRNA library index file using Bowtie to assign a matching gene-specific sgRNA, and total read count tables were subsequently generated using Samtools. A pseudocount of 1 was added to each sgRNA read count, and reads were normalized to the total read count per experimental replicate. Any sgRNA that had fewer than 25 total reads in any replicate or which were represented by less than 3 unique sgRNA for a given gene was dropped from the analysis. Average Log2 fold change was calculated for a given gene between the initial and final abundances for all sgRNAs targeting it across the replicates.

## Phylogenetic profiling analysis

To identify genes co-evolved with SHLD2, we used normalized phylogenetic profiling as previously described (Tabach *et al*, 2013a,b). Briefly, we have generated the phylogenetic profile of 42 mammalian species and calculated the Pearson correlation coefficients between the phylogenetic profile of SHLD2 and the phylogenetic profiles of 19,520 human protein coding genes, and defined the 200 genes with the highest correlation coefficients as co-evolved with SHLD2 in mammalians. In a similar manner, we identified the top 200 genes that co-evolved with SHLD2 in 63 vertebrate species. The intersection between these two lists yielded 159 genes that were subsequently considered as co-evolving with SHLD2 with high confidence and further processed for pathway enrichment analysis.

## Patient cohort analysis

TNBC patient data were collected in accordance with the McGill University Health Center research ethics board (SUR-99-780). Informed consent was obtained from all subjects, and the experiments performed are conformed to the principles set out in the WMA Declaration of Helsinki and the Department of Health and Human Services Belmont Report. Total RNA was isolated from TNBC primary tissues using Qiagen AllPrep DNA/RNA Mini Kit. RNA quality was assessed using a Bioanalyser (Agilent). RNA-Seq library was generated using the Illumina TruSeq RNA Library Prep Kit, and sequencing was performed on the Illumina HiSeq 2500 platform using 75 base-pair paired-end reads. Reads were mapped to human genome version hg19 using STAR (Spliced Transcripts Alignment to a Reference). The data were mean-centered

and log-transformed, and the expression values for *SHLD2* were extracted for further analysis. Survival analysis was performed using the coxph function in the R package "survival". Expression was classified as either low or high using the top quartile as the threshold.

### Protein purification

Recombinant SHLD2 proteins fused with a cleavable N-terminal GST tag and a C-terminal histidine tag were purified from baculovirus-infected Sf9 cells. Recombinant baculoviruses were produced by the Bac-to-Bac expression system (Invitrogen); Sf9 insect cells were infected with the different baculoviruses for 3 days at 27°C. The cells were harvested by centrifugation, and the cell pellet was resuspended in 40 ml GST buffer (PBS 1×, 150 mM KCl, 1% Triton X-100, 0.5 mM DTT, 0.019 UIT/ml Aprotinin, 1 μg/ml Leupeptin). The suspension was sonicated, and insoluble material was removed by centrifugation. Glutathione sepharose beads (GE Healthcare) were added to the supernatant and incubated for 2 h at 4°C. The beads were washed four times with GST buffer and two times with PreScission washing buffer [50 mM Tris–HCl pH 7.4, 150 mM NaCl, 1 mM EDTA, 1 mM DTT, 0.05% (v/v) Tween-20]. The proteins were eluted by cleavage with PreScission protease (80 U/ml, GE Healthcare) overnight at 4°C. The supernatant was dialyzed against Talon buffer [50 mM $NaHPO_4$ pH 7, 500 mM NaCl, 10% (v/v) glycerol, 0.05% (v/v) Triton X-100, 5 mM imidazole]. Talon beads (Clontech) were added to the supernatant and incubated for 60 min at 4°C. The resin was washed three times with Talon washing buffer (50 mM $NaHPO_4$ pH 7, 500 mM NaCl, 10% glycerol, 0.05% Triton X-100, 30 mM imidazole). SHLD2 proteins were eluted in Talon buffer containing 500 mM imidazole and dialyzed in storage buffer (20 mM Tris–HCl pH 7.5, 10% glycerol, 0.05% Tween-20, 200 mM NaCl, 1 mM DTT).

### DNA substrates and DNA-binding assays

DNA substrates used were generated with purified oligonucleotides (JYM696  GGGCGAATTGGGCCCGACGTCGCATGCTCCTCTAGACTC GAGGAATTCGGTACCCCGGGTTCGAAATCGATAAGCTTACAGTCT CCATTTAAAGGACAAG and JYM698 CTTGTCCTTTAAATGGAGAC TGTAAGCTTATCGATTTCGAACCCGGGGTACCGAATTCCTCGAGTC TAGAGGAGCATGCGACGTCGGGCCCAATTCGCCC). Briefly, double-strand DNA was prepared by annealing reaction carried out by slowly cooling from 95 to 12°C. The DNA-binding reactions (10 μl) contained $^{32}$P-labeled DNA oligonucleotides (50 nM nucleotides of each substrate) and the indicated concentrations of SHLD2 full-length or fragments in MOPS buffer (25 mM MOPS at pH 7.0, 60 mM KCl, 0.2% Tween-20, 2 mM DTT, and 5 mM $MgCl_2$). Reaction mixtures were incubated at 37°C for 20 min and then protein–DNA complexes were fixed with 0.2% (v/v) glutaraldehyde for 20 min. The reactions were subjected to electrophoresis on an 8% TBE1X-acrylamide gel, and $^{32}$P-labeled DNA was visualized by autoradiography.

## Data availability

The mass spectrometry data from this publication have been deposited to the ProteomeXchange Consortium database (http://

proteomecentral.proteomexchange.org/cgi/GetDataset) via the MassIVE partner repository and assigned the identifier PXD010648 (MassIVE code: MSV000082676).

**Expanded View** for this article is available online.

## Acknowledgements

We are grateful to Amelie Fradet-Turcotte, Michael Witcher, Josie Ursini-Siegel, Chantal Autexier, and William Foulkes for critical reading of the manuscript; to Daniel Durocher, Anne-Claude Gingras, Jeremy Stark, Michael Witcher, and Roger Greenberg for plasmids and other reagents. We would like to specifically thank Roderick McInnes, Josie Ursini-Siegel, and Koren Mann for their constant support. JH, VL, and MK received a doctoral fellowship from the Cole Foundation. AM was supported by a post-doctoral fellowship from the Cole Foundation. ESC received a FRQS post-doctoral training scholarship. HB was supported by a doctoral training award from the FRQS (#33603). JFC is the recipient of the TRANSAT chair in Breast Cancer Research. JYM is a FRQS Chair in Genome Stability. AO is the Canada Research Chair (Tier 2) in Genome Stability and Hematological Malignancies. Work in the AO laboratory was supported by a CIHR Project Grant (#376245), a CRS Operating Grant (#21038), a Transition Grant from the Cole Foundation, and an internal Operating Fund from the Sir Mortimer B. Davis Foundation of the Jewish General Hospital. Work in the JFC laboratory was supported by a NSERC Discovery Grant (RGPIN-2016-04808). Work in the AM laboratory was supported by a NSERC Discovery Grant (#5026) and a CIHR Project Grant (#376288). "Life always offers you a second chance. It is called tomorrow!". Work in the JYM laboratory was supported by a CIHR Foundation grant.

## Author contributions

SF and JH designed, performed most of the experiments presented in this manuscript, and analyzed the data. VML designed and performed the CSR experiments and analyzed the data. AMal designed, performed the CRISPR/Cas9-based genome-wide screen and the clonogenic experiments, and analyzed the data. TM and BD performed the laser stripe micro-irradiation, and AMar designed the experiments and analyzed the data. YC purified recombinant SHLD2 and performed the *in vitro* DNA-binding assays under the supervision of J-YM. ZL generated most of the constructs used in this manuscript, designed and performed the ChIP experiments, and analyzed the data. AS designed and performed the comet assay experiments and analyzed the data. ES-C, HB, DG, CD and HKh, and EGL performed the MS experiments and analyzed the data under the supervision of J-FC. MK performed the initial characterization of SHDL2 in the FokI system. DR performed the phylogenetic analysis profiling under the supervision of YT. KKM performed the pathway analysis of SHLD2. HKu and KOK performed the RNA-seq analysis and the patient cohort analysis under the supervision of CMG. MP provided the RNA-seq data and the related patient outcome of the TNBC cohort. AO conceived the study, designed the research, provided supervision, and wrote the manuscript with input from all the other authors.

### Conflict of interest

The authors declare that they have no conflict of interest.

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
