## [Review Process File · The EMBO Journal]

SHLD2/FAM35A co-operates with REV7 to coordinate DNA double-strand break repair pathway choice

Steven Findlay, John Heath, Vincent M. Luo, Abba Malina, Théo Morin, Yan Coulombe, Billel Djerir, Zhigang Li, Arash Samiei, Estelle Simo-Cheyrou, Martin Karam, Halil Bagci, Dolev Rahat, Damien Grapton, Elise G. Lavoie, Christian Dove, Husam Khaled, Hellen Kuasne, Koren K. Mann, Kathleen Oros Klein, Celia M. Greenwood, Yuval Tabach, Morag Park, Jean-Francois Côté, Jean-Yves Masson, Alexandre Maréchal and Alexandre Orthwein.

Review timeline:

Submission date:	6 th July 2018
Editorial Decision:	18 th July 2018
Revision received:	2 nd August 2018
Accepted:	3 rd August 2018

Editor: Hartmut Vordermaier

Transaction Report:

1st Editorial Decision

18th July 2018

Thank you again for submitting your manuscript on FAM35A and C20orf196 cooperation with REV7 for our editorial consideration. We have now received the enclosed reports from two expert referees, which are overall supportive but nevertheless raise a limited number of issues that would require attention. Should you be able to swiftly and adequately address these concerns, we should be happy to consider the study further for expedited publication in The EMBO Journal.

Regarding referee 1's concerns, it will be important to either modify the epistasis argument based on Fig. 4A, or to bolster it with conclusive additional data; and to temper some of the novelty claims (and to add some more discussions and comparisons to the two previous papers in the discussion, including also the "shieldin" terminology that is becoming adopted by the whole field). Additional structure-function analyses as mentioned in this referee's last point would not be essential, but certainly valuable in case you should already have obtained such insights.

For referee 2, it will be key to add missing control data, and to better support the resection studies with at least some complementary analyses. Removing the BioID data from Figure 5A would not be essential, but moving some more of the SHLD 1 data from Fig. EV5 to the main part would be helpful. With regard to the ChIP-seq experiment in Figure 6, I agree that this should best be removed unless you should already have data to substantially strengthen this. Finally, the full results of the doxorubicin sensitivity screen in Figure 5 should be included as Expanded View data set, in order to strengthen the unique value of the current manuscript and its complementarity to the already published work.

REFEREE REPORTS

Referee #1:

Findlay et al. initially identified FAM35A by mapping out the interactome of REV7, before characterizing its role in DNA repair through a number of techniques including mass spectrometry, co-immunoprecipitation assays, and others. They continued their study by defining the interactome of FAM35A to also identify C20orf196/SHDL1 that also contributes to the repair of DSBs - again specifically involved in the choice between HR and NHEJ.

Some of the above conclusions are similar to those published by two previous papers (Gupta et al. Cell 2018 and by Tomida et al. EMBOJ 2018) however this study is possibly the most extensive of the three. The experiments that are unique to this manuscript include those illustrating that the C terminus is not required for recruitment to DNA damage sites, and showing that FAM35A acts independently of TLS. The experiments are well chosen to study FAM35A and C20orf196/SHDL1, and the subsequent conclusions are argued in a clear and coherent fashion. There is enough novel material in this publication to warrant being published in the same journal as Tomida et al., however there are instances in the manuscript where the authors' claims need either modifying or additional information is needed to justify them as they stand.

Major concerns: None.

Minor concerns: The siRNA/GFP-FAM35A experiment (Fig.4A) alone does not convince me that it is appropriate to define FAM35A acting downstream of REV7 at this point (end of page 11), as there is no evidence that there isn't a co-dependence between FAM35A and REV7 to form a stable complex. Based on this experiment alone, I would preferentially describe FAM35A to be either acting in concert with or downstream of REV7, or reference other evidence at this point to determine if FAM35A solely acts downstream of REV7.

Non-essential suggestions: Given that there are now two papers published surrounding the role of FAM35A and C20orf196/SHDL1, care should be given over the language used in this manuscript, and phrases such as 'previously undescribed factor' should only be used where appropriate.

Using the truncated FAM35A variants and the pulsed IdU/CldU could possibly be an interesting way to determine what parts of the protein are responsible for protecting against end resectioning.

Referee #2:

This manuscript reports the identification of two factors, FAM35A/SHDL2 and C20orf196/SHDL1, that interact with REV7 and promote NHEJ while antagonizing HR. A mass spectrometry approach is used to identify REV7 interacting proteins, which ultimately identified both factors. A CRISPR/Cas9 screen was also performed which identified C20orf196 as a gene required for cell survival after TOP2 poison (i.e. Doxorubicin) treatment. Several standard assays are used to show that these factors are recruited to sites of damage, interact with each other in a DNA-damage independent manner, promote DSB repair by NHEJ and are required for survival from DSB-inducing treatments including IR. This is a very timely study given that two recent papers Tomida et al EMBO 2018 and Gupta et al Cell 2018. This study is complementary to these works and provides some additional information in support of the role of FAM35A and C20orf196 in regulating DSB repair. For the most part, the work and data is of high quality and technically sound. Some issues however should be addressed before publication.

Main issues.

1. Several controls are missing. For example, there is not one western blot showing the depletion of a target protein by RNA interference, whether it be siRNA or shRNA. These controls are standard in

the field and should be included to validate the levels of depletion of the proteins for these studies.

2. The evidence that FAM35A opposes HR by limiting DNA end resection is minimal. The data in Fig. 4A appears to be the only evidence and this assay is not often used to make these claims. Some additional evidence to support this statement would strengthen this claim. For example, DNA damage signaling including ATR activation can be analyzed or RPA foci formation or FACS analysis of RPA. There are several common and standard methods in the literature that are normally required to make this claim.

3. The Bio-ID experiment for FAM35A in Fig5A seems out of place since it did not identify SHLD1. This should be moved to supplemental information.

4. The potential impact of this work would increase if the screen results from Figure 5B-C were provided in the manuscript. It appears that only the main hits are given but publishing the entire screen would be very useful to the field and would further support the impact of these studies. For example, where do FAM35A, 53BP1, RIF1 and Rev7 lie in this screen. This analysis, which wouldn't need any additional experimental evidence, would support the use of this screen to identify proteins within this pathway.

5. The data showing that C20orf196/SHDL1 is recruited to DNA damage sites and promotes NHEJ should be moved from supplemental information into a main figure (i.e. EV5.B, C and D).

6. It is unclear what the value of the ChIP-Seq data is that is presented in Figure 6. Only a rudimentary analysis is performed and no data is validated and it is unclear if these experiments were even performed more than once. Without additional analyses to validate the results and to analyze them to put them in the context of some biological significance, these results are too preliminary for publication and really do not add to the manuscript. For example, there is not one western blot showing the FAM35A antibody or its validation by any technique.

Minor issues.

1. pg. 16 By mapped - By mapping
2. The tables are not very user friendly and lack labels.

General Comments

Since our first submission to *EMBO Journal*, the nomenclature in regard to the labelling of both FAM35A and C20orf196 has been modified. To reflex these changes, we have integrated the new nomenclature and re-named FAM35A by SHLD2 and C20orf196 by SHLD1 through-out the manuscript. Therefore, our revised manuscript is now entitled “**SHLD2/FAM35A co-operates with REV7 to coordinate DNA double-strand break repair pathway choice**”.

During the revision process, we have collected additional data that solidify our initial characterization of SHLD2 (FAM35A) and SHLD1 (C20orf196) in the repair of DNA double-strand breaks (DSBs) by the Non-Homologous End Joining (NHEJ) pathway. Below are highlighted the main improvements that we incorporated in our revised manuscript (in order of appearance in the result section):

- (i) using the FokI system, we confirmed that the N-terminal of SHLD2 (1-61 amino acids) is critical for its accumulation at DSBs, while its SQ site at position S339 are dispensable for its localization at sites of damage (Fig EV3.B). Interestingly, we observed that deleting the last 185 amino acids of SHLD2 partially impairs its accumulation at DSBs, which suggests a potential contribution of the C-terminus of SHLD2 localization to DSBs.
- (ii) importantly, we provide new insight in the biochemical properties of SHLD2. We purified SHLD2 from baculovirus-infected Sf9 cells (Fig EV3.C) and tested its ability to bind both single-stranded (SS) and double-stranded (DS) DNA probes. We show in Fig EV3.D that recombinant SHLD2 has the capacity to bind both probes *in vitro*. Interestingly, loss of SHLD2 first 129 amino acids (SHLD2D1-129) does not interfere greatly with its DNA binding capacity, in contrast to what we observed with a C-terminal deletion of SHLD2 (SHLD2D130-904) (Fig EV3.D). This new work suggests that SHLD2 is composed of two distinct domains:
 - a. a N-terminal motif that promotes the recruitment of SHLD2 to DSBs;
 - b. a C-terminal DNA binding module that facilitates its accumulation at DSBs.
- (iii) we show now that depletion of SHLD2 does not interfere with the accumulation of 53BP1, RIF1 and REV7 at FokI-induced DSBs (Fig EV4.B), which further suggests that SHLD2 is a

- downstream effector of REV7 in the NHEJ pathway.
- (iv) as suggested by Reviewer #2, we further ascertain the role of SHLD2 in restricting Homologous Recombination (HR). Therefore, we monitoring the phosphorylation levels of RPA2 at position S4 and S8, which is widely used as a marker for the generation of single-stranded DNA during DNA end resection. Remarkably, depletion of SHLD2 by siRNA results in a marked increase of phospho-RPA2 levels (Fig EV5.D), consistent with our previous data suggesting that SHLD2 limits DNA end resection.
 - (v) we have extended our work in the CH12F3-2 B cells and we show now that SHLD1 is important for antibody diversification by class switching (Fig.5E). In fact, depletion of SHLD1 using two distinct shRNAs impairs significantly CSR, similarly to what we observed with REV7 and SHLD2.

Altogether, we are confident that these major improvements of our manuscript firms up the mechanism by which SHLD2 controls DSB repair pathway choice by promoting NHEJ as a downstream of REV7, while limiting HR.

Point-by-point response

We would like to thank both Reviewers for their positive and very constructive comments. Below is the point-by-point response to their remarks. A bullet point always precedes our responses.

Reviewer 1

Findlay et al. initially identified FAM35A by mapping out the interactome of REV7, before characterizing its role in DNA repair through a number techniques including mass spectrometry, co-immunoprecipitation assays, and others. They continued their study by defining the interactome of FAM35A to also identify C20orf196/SHDL1 that also contributes to the repair of DSBs - again specifically involved in the choice between HR and NHEJ.

Some of the above conclusions are similar to those published by two previous papers (Gupta et al. Cell 2018 and by Tomida et al. EMBOJ 2018) however this study is possibly the most extensive of the three. The experiments that are unique to this manuscript include those illustrating that the C terminus is not required for recruitment to DNA damage sites, and showing that FAM35A acts independently of TLS. The experiments are well chosen to study FAM35A and C20orf196/SHDL1, and the subsequent conclusions are argued in a clear and coherent fashion. There is enough novel material in this publication to warrant being published in the same journal as Tomida et al., however there are instances in the manuscript where the authors claims need either modifying or additional information is needed to justify them as they stand.

Critique:

Major concerns: None.

Minor concerns: The siRNA/GFP-FAM35A experiment (Fig.4A) alone does not convince me that it is appropriate to define FAM35A acting downstream of REV7 at this point (end of page 11), as there is no evidence that there isn't a co-dependence between FAM35A and REV7 to form a stable complex. Based on this experiment alone, I would preferentially describe FAM35A to be either acting in concert with or downstream of REV7, or reference other evidence at this point to determine if FAM35A solely acts downstream of REV7.

- We thank the Reviewer for his/her important comment. We contend that, in absence of additional genetic evidence, it is preferable to describe SHLD2 as acting in concert with REV7 in the NHEJ pathway at this point of the paper. We have clarified our interpretation of the data presented in Fig.4A accordingly. To note, we have integrated new data showing that the depletion of SHLD2 does not impair significantly the accumulation of 53BP1, RIF1 and REV7 at DSBs (Fig EV4.B), which further points toward a role of SHLD2 as a downstream effector of REV7 in the NHEJ pathway.

Non-essential suggestions: Given that there are now two papers published surrounding the role of FAM35A and

C20orf196/SHDL1, care should be given over the language used in this manuscript, and phrases such as 'previously undescribed factor' should only be used where appropriate.

- This is a legitimate point raised by the Reviewer. The initial rationale of the manuscript was written prior to the publication of Gupta et al. (Cell, 2018) and Tomida et al. (EMBO J 2018). However, we contend that our manuscript should better reflect these new developments and integrate all the studies describing the role of SHLD1 and 2 in DNA repair. We have adjusted our revised manuscript accordingly and discussed in greater details how our work integrates itself with these studies.

Using the truncated FAM35A variants and the pulsed IdU/CldU could possibly be an interesting way to determine what parts of the protein are responsible for protecting against end resectioning.

- This is an interesting point that we are keen on pursuing in the future as part of another study on the mechanism(s) by which the REV7-SHLD1/2 complex protects DNA ends against resection. As discussed in the last part of our manuscript, we predict that the REV7-SHLD1/2 complex acts, in a similar to the Shelterin complex at the telomere ends (thereby the re-naming provided by the Durocher's group of Shieldin complex), by sterically hindering the access of DNA ends to processing enzymes. A more in-depth study will provide better insight into this complex and the presence of a potential catalytic activities but we believe that it is beyond the scope of our manuscript.

Reviewer 2

This manuscript reports the identification of two factors, FAM35A/SHDL2 and C20orf196/SHDL1, that interact with REV7 and promote NHEJ while antagonizing HR. A mass spectrometry approach is used to identify REV7 interacting proteins, which ultimately identified both factors. A CRISPR/Cas9 screen was also performed which identified C20orf196 as a gene required for cell survival after TOP2 poison (i.e. Doxorubicin) treatment. Several standard assays are used to show that these factors are recruited to sites of damage, interact with each other in a DNA-damage independent manner, promote DSB repair by NHEJ and are required for survival from DSB-inducing treatments including IR. This is a very timely study given that two recent papers Tomida et al EMBO 2018 and Gupta et al Cell 2018. This study is complementary to these works and provides some additional information in support of the role of FAM35A and C20orf196 in regulating DSB repair. For the most part, the work and data is of high quality and technically sound. Some issues however should be addressed before publication.

Critique:

Major issues:

1. Several controls are missing. For example, there is not one western blot showing the depletion of a target protein by RNA interference, whether it be siRNA or shRNA. These controls are standard in the field and should be included to validate the levels of depletion of the proteins for these studies.

- The Reviewer is totally right in requesting depletion efficiency obtained by RNA interference in the different experimental set-up presented in our manuscript. Due to a lack of space, we did not incorporate these data in our initial manuscript. However, we systematically assessed the knock-down efficiency obtained either by siRNA transfection or transduction with a shRNA using quantitative PCR, in absence of any commercially available antibody recognizing SHLD2 specifically. We have incorporated these data in the different supplementary figures of our revised manuscript (Fig EV2.A and C; Fig EV4.C and D; Fig EV6.D). We have also determined the knock-down efficiency of the different DNA repair factors (53BP1, RIF1, REV7 and BRCA1) by immunofluorescence using the FokI system (Fig EV4.A). Finally, we have ensured that the different shRNAs used in our CSR assay did not impact either AID expression or cell proliferation and we have incorporated these data in the corresponding supplementary figure (Fig EV4.D and E).

2. The evidence that FAM35A opposes HR by limiting DNA end resection is minimal. The data in Fig. 4A appears to be the only evidence and this assay is not often used to make these claims. Some additional evidence to support this statement would strengthen this claim. For example, DNA damage signaling including ATR activation can be analyzed or RPA foci formation or FACS analysis of RPA. There are several common and standard methods in the literature that are normally required to make this claim.

- We would like first to respectfully correct the comment made by the Reviewer, as we believe that he/she is referring to the data presented in Fig.4F and not A. Several reports have recently used modified versions of the DNA combing assay to efficiently and quantitatively monitor DNA end resection (Cruz-Garcia et al. PMID: 25310973; Lopez-Saavedra et al. PMID: 27503537; Soria-Bretones et al. PMID: 28740167; Huertas and Cruz-Garcia. PMID: 29043623). Still, we contend with the Reviewer that additional evidence would strengthen our claim that SHLD2 limits DNA end resection. Therefore, we monitored by WB the levels of phosphorylated RPA2 at positions S4/8, which is a conventional marker of DNA end resection, in U2OS cells depleted by an siRNA targeting either CtIP (which promotes DNA end resection), SHLD2 or a scrambled control (Fig EV5.D). We observed that, while depletion of CtIP reduces the levels of p-RPA following induction of DNA damage (+NCS), reduction of SHLD2 increases p-RPA levels, further confirming that SHLD2 opposes HR by limiting DNA end resection. This additional piece of evidence has been incorporated in the revised version of our manuscript.

3. The Bio-ID experiment for FAM35A in Fig5A seems out of place since it did not identify SHLD1. This should be moved to supplemental information.

- We contend with the Reviewer that the relevance of the BioID as part of the main Figure is questionable based on the absence of SHLD1 as a high-confidence interactor and we have moved these data to Fig EV6.B.

4. The potential impact of this work would increase if the screen results from Figure 5B-C were provided in the manuscript. It appears that only the main hits are given but publishing the entire screen would be very useful to the field and would further support the impact of these studies. For example, where do FAM35A, 53BP1, RIF1 and Rev7 lie in this screen. This analysis, which wouldn't need any additional experimental evidence, would support the use of this screen to identify proteins within this pathway.

- We thank the Reviewer for his/her important comment and we fully agree that sharing the entire data related to our screen will be useful to the DNA repair field. The complete screen has been incorporated as a separate Table where the score for each gene has been indicated (Table 3). As suggested by the Reviewer, we indicated in Fig.5B the score of 53BP1, RIF1 and REV7 in regards to their sensitivity to Doxorubicin. Unfortunately, SHLD2 was not initially incorporated in the design of the TKO.v1 library and could therefore not be analyzed for its sensitivity to Doxorubicin in our genome-wide approach.

5. The data showing that C20orf196/SHDL1 is recruited to DNA damage sites and promotes NHEJ should be moved from supplemental information into a main figure (i.e. EV5.B, C and D).

- We agree with the Reviewer and we have moved the data describing the role of SHLD1 as a key factor in the NHEJ pathway in the main figure (Fig.5C, D and F). Furthermore, we incorporated our new data showing that depletion of SHLD1 impairs CSR in CH12F3-2 B-cells (Fig.5E). The data showing that SHLD1 is recruited to DNA damage sites have been kept in the supplemental information due to a lack of space (Fig EV6.C)

6. It is unclear what the value of the ChIP-Seq data is that is presented in Figure 6. Only a rudimentary analysis is performed and no data is validated and it is unclear if these experiments were even performed more than once. Without additional analyses to validate the results and to analyze them to put them in the context of some biological

significance, these results are too preliminary for publication and really do not add to the manuscript. For example, there is not one western blot showing the FAM35A antibody or its validation by any technique.

- We contend with the Reviewer that our ChIP-Seq data are not the strongest data presented in this manuscript and too preliminary for publication. Therefore, we have removed them in the revised manuscript.

Minor issues:

1. pg. 16 By mapped - By mapping

- We have modified accordingly the text based on the Reviewer's comment.

2. The tables are not very user friendly and lack labels.

- We agree with the Reviewer that our tables could be more user friendly and we have modified them accordingly.

Accepted

3rd August 2018

Thank you for submitting your final revised manuscript for our consideration. I have now had chance to check your responses to the original comments and to assess the newly added data, and I am pleased to inform you that following this we have now accepted it for publication in The EMBO Journal.

Corresponding Author Name: Alexandre Orthwein

Manuscript Number: EMBOJ-2018-100158